# Following the Navigation: Enhancing Small Language Models contextual reasoning with LLM Guidance

**Xiaoqi Ni[1], Jie Wang[1*], Lin Yang[2], Yiyang Lu[1], Hanzhu Chen[1], Rui Liu[1], Jianye Hao[3]**
[1]MoE Key Laboratory of Brain-inspired Intelligent Perception and Cognition,
University of Science and Technology of China
[2]Huawei Technologies Co., Ltd.
[3]College of Intelligence and Computing, Tianjin University
`xiaoqi_ni@mail.ustc.edu.cn`

## Abstract

Large language models (LLMs), such as OpenAI o1 and DeepSeek-R1, excel in contextual reasoning by leveraging extensive world knowledge and deep contextual understanding. However, their high computational costs limit deployment in resource-constrained settings. Conversely, small language models (SLMs) are more computationally efficient but often struggle with contextual reasoning due to limited parameter capacity and challenges like catastrophic forgetting. Existing enhancement methods for SLMs—such as knowledge distillation and data synthesis—still depend on additional training and face inherent limitations. To address this, we propose **Navigation**, a novel training-free framework that improves SLMs' contextual reasoning by distilling LLM-derived contextual processing expertise into generalizable Navigation templates. These templates, stored in a scalable Navigation database, guide SLMs through a three-stage process—**Generation**, **Utilization**, and **Update**—to locate and process critical information within complex contexts. Experiments demonstrate that our approach yields an average **10.7%** accuracy gain with a template count equivalent to no more than **2.1%** of the dataset size, enabling models such as Qwen2.5-3B-Instruct and Llama-3.2-3B-Instruct to outperform GPT-3.5-Turbo on diverse contextual reasoning tasks.

## 1 Introduction

Currently, large language models (LLMs) have demonstrated remarkable capabilities in various tasks, including multilingual generation and comprehension, coding, mathematics, and symbolic reasoning (Dubey et al., 2024; Yang et al., 2025a; Comanici et al., 2025; Bai et al., 2026). Among these, contextual reasoning is a key ability. These tasks require extensive world knowledge and often demand careful processing and a deep understanding of context (Petroni et al., 2020). This requires LLMs to comprehend and extract key information from texts, identify logical relationships and dependencies between different parts, and make accurate inferences and responses. While LLMs, such as OpenAI o1 (Jaech et al., 2024) and DeepSeek-R1 (Guo et al., 2025), have excelled in processing and understanding complex contextual information, scaling these models involves significant computational and deployment costs. This has led to a renewed interest in small, resource-efficient models, such as those with 3B parameters, which can be deployed on edge devices (Chen & Varoquaux, 2024). Despite their appeal, these small models face critical limitations in contextual understanding and complex reasoning due to their constrained ability to model complex dependencies (Zhu et al., 2024).

To address this challenge, existing work has focused on training-based methods to transfer emergent abilities (Wei et al., 2022a) of large models to small models. These methods primarily include knowledge distillation and data synthesis. In **knowledge distillation**, a compact "student" model mimics a

---

*Corresponding author. Email: jiewangx@ustc.edu.cn.

larger "teacher" model. White-box distillation uses internal representations like logits and hidden states to create efficient models, e.g., DistilBERT (Sanh et al., 2019) and QuantizedGPT (Yao et al., 2022). Black-box distillation employs teacher-generated datasets with pseudo-labels or reasoning traces, as in CoT (Wei et al., 2022b) and Instruction-Following Distillation (Hsieh et al., 2023), boosting SLM reasoning and zero-shot generalization. **Data synthesis** uses LLMs to generate or augment training data, such as synthetic datasets or paraphrased examples (Meng et al., 2022; Ma et al., 2023), enhancing SLM performance in tasks like intent classification and dialogue understanding (Tan et al., 2024). Despite these advances, **SLMs still struggle with contextual reasoning tasks due to limited parameter capacity and catastrophic forgetting** (Liu et al., 2023), underscoring the importance of **effective data memorization** for language task performance (Brown et al., 2021).

To overcome these limitations, we propose **Navigation**, a novel training-free framework designed to enhance small language models (SLMs) for contextual reasoning tasks. Navigation provides generalizable templates that guide SLMs to effectively locate and utilize critical information within complex, information-dense contexts, preventing them from becoming overwhelmed. These structured templates, distilled from large language models (LLMs), are stored in a Navigation database. By enabling SLMs to leverage LLM-derived contextual processing expertise without exceeding their limited capacity, Navigation mitigates issues like catastrophic forgetting and enhances overall reasoning accuracy. There is no universally accepted definition distinguishing large models from small ones (Chen & Varoquaux, 2024). In this work, we consider model size in relative terms. For example, Llama-3.2-3B-Instruct (Dubey et al., 2024) is considered small compared to DeepSeek-R1 (671B parameters) (Guo et al., 2025) and GPT-3.5-Turbo (175B parameters)[1].

The Navigation framework operates through a three-stage process to empower SLMs in contextual reasoning tasks. In the **Navigation Generation** stage, LLMs identify and abstract critical information while discarding irrelevant details, formulating clear and concise general reasoning guidance that is both comprehensible and adaptable for small models, ensuring broader applicability across similar tasks. This guidance, along with its corresponding problems, is structured into Navigation templates and stored in a Navigation database for efficient retrieval and application. During the **Navigation Utilization** phase, the most relevant template is retrieved by matching the task query to Navigation templates through semantic similarity. SLMs then follow the template's guidance step-by-step to extract key information, forming a reasoning chain that reduces interference from irrelevant details in complex contexts. The **Navigation Update** mechanism enhances scalability by identifying gaps in template coverage, prompting LLMs to generate or refine templates for novel or underrepresented tasks, thus dynamically expanding the database.

Our main contributions are summarized as follows:

- **A Novel Training-Free Reasoning Framework.** We propose **Navigation**, a framework that empowers Small Language Models (SLMs) by distilling abstract *reasoning strategies*—not just textual outputs—from powerful LLMs. These strategies are operationalized as reusable Navigation templates that guide SLMs to overcome contextual distractions. This approach enables a 3B parameter model to achieve an average **10.7% accuracy gain** and surpass the performance of the much larger GPT-3.5-Turbo.

- **An Adaptive and Resource-Efficient Guidance Mechanism.** The Navigation framework operates via a dynamic three-stage process: **Generation, Utilization, and Update**. This allows the system to scalably adapt to new tasks by generating new templates on-the-fly. The mechanism is highly resource-efficient, requiring a template database amounting to no more than **2.1% of the dataset size** to achieve these significant performance gains, a fraction of the cost of other LLM-guidance methods.

- **Rigorous and Comprehensive Empirical Validation.** We provide a rigorous validation across **3 diverse reasoning benchmarks**, encompassing a total of **8 metrics** (including Accuracy, F1, and EM). Across these extensive evaluations, SLMs (from **3B to 7B** parameters) equipped with Navigation consistently outperform strong training-based baselines. Furthermore, our qualitative analysis reveals that Navigation improves the *quality* of reasoning, helping models follow logical causality over superficial contextual cues.

---

[1] https://openai.com. 175B is an estimated value.

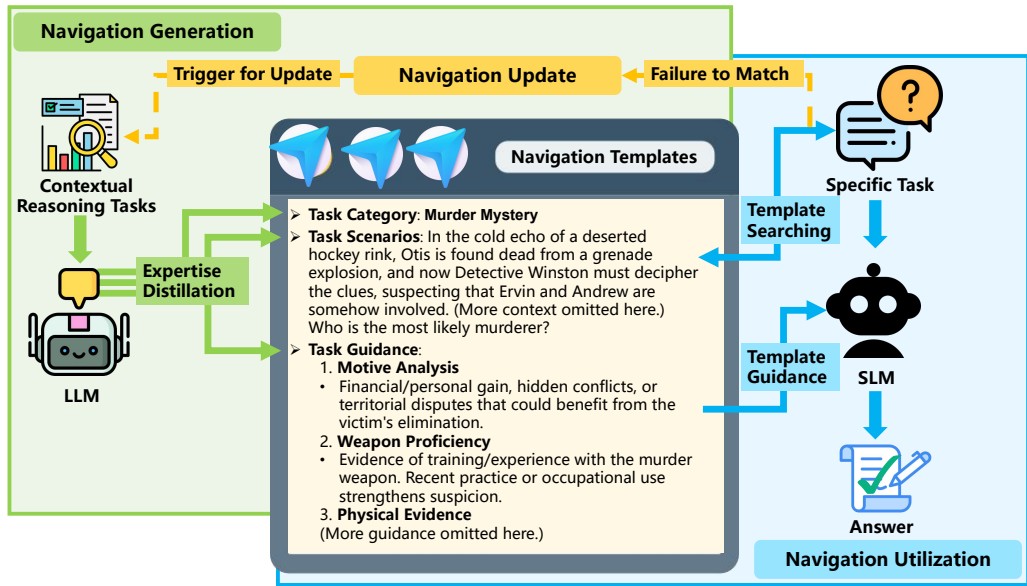

Figure 1: Overview of the Navigation framework.

# 2  PRELIMINARIES

## 2.1  ADDRESSING CONTEXTUAL LIMITATIONS THROUGH ROBUST IN-CONTEXT LEARNING

Small language models have demonstrated notable robustness in in-context learning (ICL) under noisy conditions, effectively prioritizing salient features while suppressing irrelevant information (Wei et al., 2023). Nevertheless, their performance remains highly sensitive to prompt quality. We posit that strategically engineered prompts can exploit this inherent robustness to mitigate contextual understanding limitations, thereby enhancing reasoning performance without resource-intensive scaling or fine-tuning (Liu et al., 2023).

## 2.2  ADDRESSING MEMORY CONSTRAINTS WITH EXTERNAL KNOWLEDGE BASES

Small language models (SLMs) suffer from limited capacity and catastrophic forgetting, which hinder their contextual reasoning compared to large language models (LLMs) leveraging vast memorized data (Carlini et al., 2022; Tirumala et al., 2022). External knowledge bases (KBs) with non-parametric retrieval can alleviate these issues by reducing memory demands and mitigating forgetting, often outperforming knowledge distillation (Brown et al., 2021; Kang et al., 2023). We hypothesize that a specialized KB, Navigation, can further distill LLM-style contextual processing to enhance SLMs' performance under memory constraints (see Appendix B for theoretical details).

# 3  METHOD

In this section, we introduce **Navigation**, a novel framework that, with large language model (LLM) guidance, prevents small models from getting lost in the middle of information-dense contexts, enabling them to comprehensively capture critical contextual information and reason accurately. An overview of Navigation is shown in Figure 1, comprising three main steps: **Generation**, **Utilization**, and **Update**. Firstly, Navigation stores LLM contextual processing expertise (Section 3.1). Next, for a given query and context, Navigation retrieves the most relevant guidance template based on the current scenario, allowing small models to instantiate and analyze key points step-by-step for robust reasoning (Section 3.2). Finally, to enhance Navigation's scalability, the Navigation database is dynamically updated as new tasks emerge (Section 3.3).

### 3.1 Navigation Generation

Navigation Generation is designed to distill the contextual processing expertise of LLMs into structured, reusable guidance that enables small models to locate and utilize critical information effectively.

**Generalizable Guidance.** Different types of reasoning tasks require distinct contextual information, particularly those demanding expert knowledge to ensure a complete and rigorous reasoning chain. For example, in a murder mystery inference task, a suspect's motives, means, and opportunities must all be thoroughly evaluated; without specialized domain knowledge, models may erroneously identify the murderer by focusing on only one or two factors. In contrast, a team task assignment problem demands a comprehensive analysis of candidates' strengths, weaknesses, and interpersonal dynamics to achieve optimal outcomes. This variability necessitates tailored guidance for small language models (SLMs), which struggle with complex contextual dependencies due to limited capacity.

**Guidance Distillation.** To address this challenge, Navigation leverages the expertise of large language models (LLMs) to provide precise, generalizable guidance. The process begins with a complete reasoning cycle, encompassing problem comprehension, text parsing, information extraction, and the construction of a reasoning chain to produce an answer. During this cycle, the LLM identifies and abstracts critical information relevant to the task, filtering out irrelevant details to minimize noise. The distilled knowledge is then organized into structured Navigation templates, categorized by problem type, and linked to generalizable directives. This ensures that SLMs receive clear, tailored instructions rather than generic guidance, enabling effective performance in contextual reasoning tasks despite their resource constraints. The Navigation generation prompts are included in Appendix C.1.

**Navigation Structuring.** The distilled knowledge from the LLM's reasoning cycle is systematically organized into a structured Navigation template to provide actionable, generalizable guidance for SLMs (in Appendix C.2). Each template comprises the following components:

- **Task Category**: Identifies the problem type (e.g., "Murder Mystery" or "Team Assignment") to distinguish tasks requiring specialized expertise, enabling targeted retrieval of relevant guidance.
- **Task Scenarios**: Archives queries and contexts from prior LLM-processed tasks, allowing the Navigation system to select the most appropriate guidance template through similarity-based matching.
- **Task Guidance**: Enumerates critical abstract information types (e.g., "Alibi Credibility" or "Skill-to-Role Alignment") and provides concise, generalizable explanations to facilitate precise information retrieval and application (e.g., "Alibi Credibility: Corroborated presence at another location during the crime via witnesses, digital records, or timestamped activities.").

### 3.2 Navigation Utilization

The Navigation Utilization phase empowers small models to effectively leverage structured Navigation templates to locate critical information and perform robust reasoning. By retrieving and applying generalizable guidance, this phase enables small language models (SLMs) to navigate complex, information-dense contexts with precision, overcoming their inherent limitations in processing capacity.

**Template Matching.** Upon receiving a task query and its associated context, an additional embedding model identifies the most relevant Navigation template by computing cosine similarity to match the current task scenario against those in the Navigation database. Crucially, the embedding model operates independently of both the SLM and LLM architecture. This is accomplished by computing the semantic similarity between the vector representation of the current task scenario, $f(x_d)$, and the vector representations of the Task Scenarios stored in the database, $\{f(D_{T_i})\}_{i=1}^N$. The matching process is formalized as:

$$j = \arg\max_i \text{Sim}(f(x_d), f(D_{T_i})), \tag{1}$$

where $\text{Sim}(\cdot, \cdot)$ denotes a similarity metric (e.g., cosine similarity), $f(\cdot)$ represents a standard embedding model, $x_d$ is the current task scenario (comprising the query and context), and $\{D_{T_i}\}_{i=1}^N$ are

the Task Scenarios archived in the Navigation database. If the similarity score exceeds a predefined threshold $\delta$, the corresponding template is selected to guide the SLM's reasoning process. If no template meets the threshold, the system triggers the Navigation Update mechanism (Section 3.3) to generate a new template.

**Information Localization and Reasoning.** Using the selected Navigation template, the small model systematically scans the input text to identify and extract task-relevant information corresponding to the template's directives. For example, if the template specifies "Alibi Credibility", the model targets evidence such as witness statements, digital records, or timestamped activities, filtering out irrelevant details to prevent information overload. The extracted information is then integrated to construct a structured reasoning chain, enabling the model to generate a concise and accurate response. This is an end-to-end process where the SLM performs both template instantiation and inference within a single generation step, which not only significantly boosts efficiency but also reduces the likelihood of hallucination. By focusing on pre-identified critical details, the SLM avoids redundant processing, ensuring efficient and robust task completion.

## 3.3 NAVIGATION UPDATE

The Navigation Update mechanism ensures the framework's adaptability and scalability by dynamically expanding its coverage to address new or underrepresented task types, maintaining robustness in dynamic, real-world applications.

**Continuous Learning.** When a small model fails to retrieve a suitable Navigation template (i.e., the similarity score $\text{Sim}(f(x_d), f(D_{T_i})) < \delta$) or an existing template inadequately guides information localization and reasoning, a Navigation gap is detected. The small model logs the problem type, task characteristics, and failure reasons (e.g., "missing guidance for novel task type"), and this information is passed to the large language model (LLM). When the existing templates in the Navigation database are insufficient to guide the SLM in answering new questions, the LLM is invoked to generate appropriate reasoning templates. The LLM identifies the question type, serving as a label for subsequent management, and simultaneously generates corresponding general guidance. This guidance assists the SLM in locating key contextual information and performing inference. Importantly, the LLM does not provide the final answer, ensuring the small model remains responsible for task execution. Newly generated or updated templates are stored in the Navigation database, either replacing outdated entries or supplementing the repository. Through this continuous learning process, the database evolves dynamically as new tasks are encountered, enhancing the framework's generalization and adaptability across diverse and evolving task domains.

**Real-World Operational Workflow.** Our methodology is designed for real-world scenarios that prioritize operational efficiency. The system first attempts to resolve user queries using the SLM. If the template matching mechanism fails to identify a relevant template, the query is routed to the LLM. Upon generating a response, the LLM identifies categories of key information essential for reasoning to formulate a new template, thereby dynamically updating the database. This allows future questions of a similar nature to be handled by the SLM using the newly acquired template. This mechanism aims to maximize the SLM's coverage while minimizing computationally expensive LLM calls.

## 4 EXPERIMENTS

### 4.1 EXPERIMENTAL SETTINGS

**Evaluation Benchmarks.** To evaluate the effectiveness of our approach, we focus on contextual reasoning tasks requiring extensive world knowledge, precise contextual interpretation, and multi-step reasoning (Petroni et al., 2020). Our primary benchmark is **MuSR** (Sprague et al., 2023), a narrative-driven benchmark for multi-step commonsense reasoning. MuSR covers three domains—**Object Placements** (OP), **Murder Mystery** (MM), and **Team Allocation** (TA)—with about 250 questions each that demand evidence retrieval and stepwise reasoning to reach the correct answer. To further assess generalizability, we utilize **StrategyQA** (Geva et al., 2021), comprising 2,061 yes/no questions requiring retrieval and reasoning over supporting evidence across diverse domains. In addition, we incorporate **HotpotQA** (Yang et al., 2018), a large-scale Wikipedia-based benchmark explicitly designed to test explainable multi-hop reasoning over multiple documents. For all benchmarks, we

Table 1: Main results on contextual reasoning benchmarks, including the Object Placements (OP), Murder Mystery (MM), and Team Allocation (TA) domains from the MuSR (Sprague et al., 2023) dataset, as well as the StrategyQA (Geva et al., 2021) and HotpotQA (Yang et al., 2018) datasets. $\triangle$ denotes the margin between Vanilla and Navigation. "EM" indicates the exact match of HotpotQA. Navigation (DS-R1) and Navigation (GPT-5.1) denote Navigation templates generated via DeepSeek-R1 and GPT-5.1, respectively. We bold the best results for each SLM backbone and underline the second-best results.

| Models | MuSR | | | StrategyQA | | | HotpotQA |
|---|---|---|---|---|---|---|---|
| | OP | MM | TA | Accuracy | F1 | Recall | EM |
| **Large Language Models** | | | | | | | |
| **GPT-3.5-Turbo** | $44.6_{\pm 0.5}$ | $60.3_{\pm 0.9}$ | $42.4_{\pm 1.4}$ | $68.1_{\pm 0.4}$ | $61.1_{\pm 0.5}$ | $53.5_{\pm 0.6}$ | $44.4_{\pm 0.2}$ |
| **Llama-3.3-70B-Instruct** | $42.7_{\pm 0.2}$ | $63.6_{\pm 0.0}$ | $68.4_{\pm 0.0}$ | $80.6_{\pm 0.0}$ | $76.3_{\pm 0.0}$ | $66.5_{\pm 0.0}$ | $59.9_{\pm 0.0}$ |
| **DeepSeek-R1** | $55.3_{\pm 1.6}$ | $73.5_{\pm 0.8}$ | $84.5_{\pm 1.5}$ | $82.0_{\pm 0.6}$ | $77.9_{\pm 0.9}$ | $67.7_{\pm 1.0}$ | $52.8_{\pm 0.7}$ |
| **Small Language Models** | | | | | | | |
| **Qwen2.5-3B-Instruct** | | | | | | | |
| **Vanilla** | $41.0_{\pm 0.0}$ | $55.6_{\pm 0.0}$ | $34.5_{\pm 1.2}$ | $59.6_{\pm 0.0}$ | $27.5_{\pm 0.0}$ | $16.4_{\pm 0.0}$ | $34.9_{\pm 0.0}$ |
| **+ COT** | $45.2_{\pm 0.5}$ | $57.7_{\pm 0.5}$ | $40.1_{\pm 2.3}$ | $62.5_{\pm 0.0}$ | $40.3_{\pm 0.0}$ | $27.1_{\pm 0.0}$ | $39.6_{\pm 0.6}$ |
| **+ SLEICL** | $51.2_{\pm 8.1}$ | $58.7_{\pm 2.4}$ | $40.5_{\pm 4.8}$ | $60.9_{\pm 1.9}$ | $18.4_{\pm 1.1}$ | $10.3_{\pm 0.7}$ | $37.1_{\pm 0.0}$ |
| **+ SFT** | $34.6_{\pm 0.0}$ | $58.7_{\pm 2.3}$ | $\mathbf{48.0}_{\pm 0.0}$ | $60.9_{\pm 0.0}$ | $36.2_{\pm 0.0}$ | $24.0_{\pm 0.0}$ | $37.7_{\pm 0.6}$ |
| **+ Navigation (DS-R1)** | $\underline{52.6}_{\pm 0.0}$ | $\underline{60.5}_{\pm 0.6}$ | $44.6_{\pm 0.5}$ | $\mathbf{66.4}_{\pm 0.0}$ | $\mathbf{53.5}_{\pm 0.0}$ | $\underline{40.6}_{\pm 0.0}$ | $\underline{51.1}_{\pm 0.1}$ |
| **+ Navigation (GPT-5.1)** | $\mathbf{52.7}_{\pm 0.2}$ | $\mathbf{64.5}_{\pm 0.0}$ | $\underline{45.0}_{\pm 0.2}$ | $\underline{65.9}_{\pm 0.0}$ | $\underline{53.4}_{\pm 0.0}$ | $\mathbf{41.8}_{\pm 0.0}$ | $\mathbf{51.8}_{\pm 0.0}$ |
| $\triangle$ | +11.7 | +8.9 | +10.5 | +6.8 | +26.0 | +25.4 | +16.9 |
| **Llama-3.2-3B-Instruct** | | | | | | | |
| **Vanilla** | $43.4_{\pm 1.0}$ | $53.2_{\pm 0.7}$ | $44.9_{\pm 0.2}$ | $59.9_{\pm 0.0}$ | $28.2_{\pm 0.2}$ | $16.9_{\pm 0.2}$ | $35.8_{\pm 0.2}$ |
| **+ COT** | $47.2_{\pm 0.5}$ | $54.7_{\pm 0.2}$ | $45.9_{\pm 0.5}$ | $61.4_{\pm 0.1}$ | $34.1_{\pm 0.2}$ | $21.4_{\pm 0.1}$ | $37.8_{\pm 1.2}$ |
| **+ SLEICL** | $38.4_{\pm 4.2}$ | $50.8_{\pm 1.8}$ | $25.0_{\pm 5.2}$ | $60.8_{\pm 0.9}$ | $54.9_{\pm 1.0}$ | $\mathbf{55.4}_{\pm 0.7}$ | $37.0_{\pm 2.6}$ |
| **+ SFT** | $38.5_{\pm 0.0}$ | $52.0_{\pm 0.0}$ | $44.0_{\pm 0.0}$ | $63.3_{\pm 0.5}$ | $38.4_{\pm 0.9}$ | $24.7_{\pm 0.6}$ | $35.0_{\pm 1.7}$ |
| **+ Navigation (DS-R1)** | $\underline{52.7}_{\pm 0.2}$ | $\underline{63.1}_{\pm 0.0}$ | $\underline{47.1}_{\pm 0.5}$ | $\underline{68.7}_{\pm 0.0}$ | $\underline{60.8}_{\pm 0.0}$ | $51.1_{\pm 0.0}$ | $\underline{51.9}_{\pm 0.5}$ |
| **+ Navigation (GPT-5.1)** | $\mathbf{53.5}_{\pm 0.5}$ | $\mathbf{64.5}_{\pm 0.0}$ | $\mathbf{48.5}_{\pm 0.6}$ | $\mathbf{69.4}_{\pm 0.0}$ | $\mathbf{61.7}_{\pm 0.0}$ | $\underline{52.9}_{\pm 0.0}$ | $\mathbf{53.3}_{\pm 0.3}$ |
| $\triangle$ | +10.1 | +11.3 | +3.6 | +9.5 | +33.5 | +36.0 | +17.5 |
| **Qwen2.5-7B-Instruct** | | | | | | | |
| **Vanilla** | $47.3_{\pm 0.4}$ | $54.4_{\pm 0.4}$ | $39.5_{\pm 1.3}$ | $68.5_{\pm 0.0}$ | $53.1_{\pm 0.0}$ | $38.2_{\pm 0.0}$ | $41.8_{\pm 0.0}$ |
| **+ COT** | $51.3_{\pm 0.2}$ | $58.0_{\pm 0.0}$ | $41.7_{\pm 0.9}$ | $72.0_{\pm 0.6}$ | $62.9_{\pm 0.9}$ | $50.7_{\pm 0.9}$ | $47.6_{\pm 0.5}$ |
| **+ SLEICL** | $54.6_{\pm 10.3}$ | $60.3_{\pm 0.7}$ | $36.9_{\pm 2.1}$ | $67.5_{\pm 1.9}$ | $42.4_{\pm 1.2}$ | $27.8_{\pm 1.1}$ | $44.7_{\pm 0.2}$ |
| **+ SFT** | $26.9_{\pm 0.0}$ | $56.0_{\pm 0.0}$ | $41.3_{\pm 2.3}$ | $\mathbf{77.0}_{\pm 0.3}$ | $\mathbf{69.5}_{\pm 0.5}$ | $\mathbf{56.6}_{\pm 0.6}$ | $43.3_{\pm 1.2}$ |
| **+ Navigation (DS-R1)** | $\underline{57.6}_{\pm 0.5}$ | $\underline{63.8}_{\pm 0.6}$ | $\mathbf{47.8}_{\pm 0.4}$ | $\underline{74.6}_{\pm 0.0}$ | $\underline{68.4}_{\pm 0.0}$ | $\underline{57.2}_{\pm 0.0}$ | $\mathbf{52.3}_{\pm 0.1}$ |
| **+ Navigation (GPT-5.1)** | $\mathbf{60.8}_{\pm 0.2}$ | $\mathbf{66.1}_{\pm 0.0}$ | $46.3_{\pm 0.2}$ | $74.2_{\pm 0.0}$ | $66.7_{\pm 0.0}$ | $55.2_{\pm 0.0}$ | $\underline{52.0}_{\pm 0.0}$ |
| $\triangle$ | +13.5 | +11.7 | +8.3 | +6.1 | +15.3 | +19.0 | +10.5 |

report performance using **accuracy**, **F1 score**, **recall**, or **exact match** (EM), as appropriate for each dataset.

**Implementation and Baselines.** For small language models (SLMs), we prioritize the latest models with parameter sizes between 3 billion and 7 billion, including **Qwen (3B, version 2.5)** (Yang et al., 2024a), **Llama (3B, version 3.2)** (Dubey et al., 2024), and **Qwen (7B, version 2.5)** (Yang et al., 2024a). For large language models (LLMs), we select **GPT-3.5-Turbo (175B)** and **Llama (70B, version 3.3)** (Dubey et al., 2024) to serve as **reference models**, while employing **DeepSeek-R1 (671B)** (Guo et al., 2025) and **GPT-5.1** respectively as the LLMs for **template generation**. For both LLMs and SLMs, we focus on conversational versions and use each model's default template to

ensure consistency. More details on the implementation are provided in Appendix D.1. We compare our Navigation method with the following approaches:

- **Vanilla**: This is our most basic baseline, where the language model (LM) generates responses directly based on the input query, without specific input-output examples or additional instructions beyond the task description.
- **CoT**: CoT (Wei et al., 2022b) encourages LLMs to first generate internal thoughts or reasoning steps before responding.
- **SLEICL**: SLEICL (Strong LLM Enhanced In-Context Learning) (Chen et al., 2024) leverages powerful language models to understand and rephrase specific task requirements from demonstrations as additional instructions, thereby enhancing the weaker models' In-Context Learning capability and improving their performance in classification tasks.
- **SFT**: Supervised fine-tuning (SFT) is a standard process in which a pre-trained model is further trained on a labeled dataset to adapt it to a specific downstream task. To improve efficiency, we employ LoRA (Hu et al., 2022), a parameter-efficient fine-tuning approach that updates only a small set of low-rank adaptation matrices while keeping most of the pre-trained parameters frozen.

**Simulation of Adaptive Workflow.** Our experimental setup is designed to simulate the **Real-World Operational Workflow** described in Section 3.3 and to verify the effectiveness of our framework. In each run, we reshuffle the dataset to simulate the SLM processing user queries in a random sequence. Instances where template matching fails—thereby triggering the LLM for template generation—are excluded from accuracy statistics, ensuring the fairness of our experimental evaluation.

## 4.2 MAIN RESULTS

Our experimental results, as shown in Table 1, demonstrate that our proposed Navigation method yields **consistent and significant improvements** across all backbones and benchmarks. For instance, on Qwen2.5-3B-Instruct, Navigation improves MuSR average (averaged over OP, MM, and TA) from 43.7% to 54.1% (+10.4) and HotpotQA exact match (EM) from 34.9% to 51.8% (+16.9). Similar gains are observed on Llama-3.2-3B-Instruct (+8.3 MuSR avg., +17.5 EM) and Qwen2.5-7B-Instruct (+11.2 MuSR avg., +10.5 EM), with small variances across runs.

Notably, Qwen2.5-3B-Instruct and Llama-3.2-3B-Instruct with Navigation notably surpass GPT-3.5-Turbo (175B) on MuSR and HotpotQA, while Qwen2.5-7B-Instruct with Navigation outperforms GPT-3.5-Turbo across all datasets and metrics. This highlights the **scalability and efficiency** of our approach. The consistent performance gains observed when applying Navigation templates, generated by diverse LLMs, to various SLMs further underscore the **robustness and generalizability** of our method across different model architectures and instruction styles.

We attribute our method's success to the Navigation templates, which provide step-by-step guidance distilled from large models' reasoning traces. These templates, written in concise and accessible language **tailored to the comprehension abilities of small models**, allow them to focus on critical information while effectively minimizing the inclusion of irrelevant context. In contrast, other baselines like COT, SLEICL, and SFT show limitations: COT struggles to mitigate context distractions despite activating multi-step reasoning; SLEICL offers generalizable reformulations that lead to unstable improvements and poor generalization in open-domain tasks; and SFT's reliance on large, high-quality labeled datasets limits its performance, especially on smaller datasets like MuSR. Additional main results for SOTA SLMs and other baselines are provided in Appendix D.2.

## 4.3 COST ANALYSIS

Given the typically high costs associated with LLM inference, we will analyze and discuss the inference and additional costs incurred by our proposed method in this section. Our primary objective is to **minimize reliance on LLMs** and to **reduce the overhead** associated with generating or utilizing templates.

We conducted a comprehensive evaluation of inference costs, including **latency**, **prompt tokens**, **output tokens**, and **GFLOPs**, across various methods using the MuSR dataset with Llama-3.2-3B-Instruct as the SLM, as detailed in Table 2. The integration of Navigation templates notably

Table 2: Cost statistics on the MuSR (Sprague et al., 2023) dataset, employing Llama-3.2-3B-Instruct (Dubey et al., 2024) as the SLM.

| Methods | Accuracy | Latency | Prompt Tokens | Output Tokens | GFLOPs | LLM Call Frequency | Average LLM Output Tokens | Training Time |
|---|---|---|---|---|---|---|---|---|
| Vanilla | 47.2 | 21.8 | 1010 | 6 | 6441 | – | – | – |
| + COT | 49.3 | 27.0 | 1024 | 15 | 6587 | – | – | – |
| + SLEICL | 38.1 | 432.4 | 1428 | 540 | 12477 | 502 | 1402 | – |
| + SFT | 44.8 | 100.7 | 1010 | 6 | 6460 | – | – | 6 min 55 s |
| + Navigation | 54.3 | 175.5 | 1305 | 934 | 14195 | 16 | 479.6 | – |

Table 3: Statistics of Navigation costs across different datasets using intfloat/e5-mistral-7b-instruct (Wang et al., 2023) as the embedding model, including the number of QA pairs, similarity threshold for template matching, number of Navigation templates generated, total LLM calls, and average time (in milliseconds) for template retrieval per question.

| Dataset | Size | Similarity Threshold | Template Count | LLM Call Frequency | Average Retrieval Latency (ms) |
|---|---|---|---|---|---|
| MuSR | 756 | 0.7 | 8 | 16 | 107.4 |
| StrategyQA | 2061 | 0.5 | 13 | 26 | 148.3 |
| HotpotQA | 1000 | 0.5 | 21 | 42 | 199.5 |

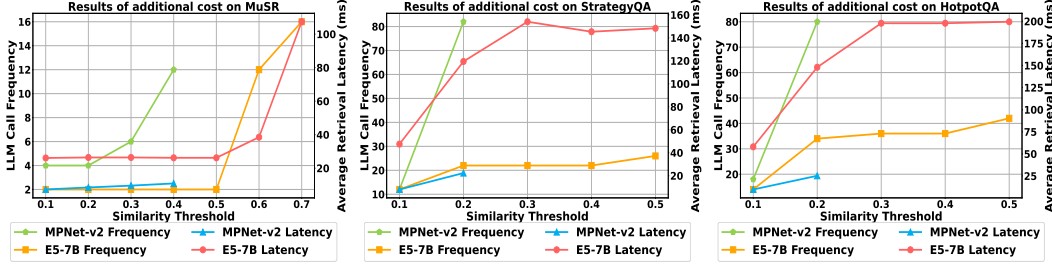

Figure 2: Results of the additional Navigation costs for sentence-transformers/all-mpnet-base-v2 (MPNet-v2) (Song et al., 2020) and intfloat/e5-mistral-7b-instruct (E5-7B) (Wang et al., 2023) across different datasets under varying thresholds. The thresholds range from 0.1 to the threshold corresponding to the maximum cost within the controllable range, with additional Navigation costs increasing monotonically as the threshold increases.

improved latency and GFLOPs. Crucially, the observed increase in output length suggests an **effective activation of the SLM's capacity for context analysis and text-based reasoning**, which in turn leads to a significant enhancement in its overall reasoning performance. Furthermore, comparing our LLM-dependent method with SLEICL, we report LLM call frequency and average output tokens for template generation. Our approach leverages LLMs to distill general and concise guidance templates, a strategy that significantly **reduced LLM call frequency** (to only 3% of SLEICL's) **and associated costs**, while simultaneously achieving **superior performance** compared to directly using few-shot examples.

As shown in Table 3, the MuSR dataset includes 8 LLM-generated templates within its 756 samples, representing approximately **1%** of the total. In comparison, StrategyQA employs 13 templates across 2,061 samples, yielding a proportion of **0.6%**. Conversely, the **SLEICL** (Chen et al., 2024) methodology exhibits a substantial reliance on LLM-generated demonstrations, which constitute **66.7%** of its total dataset. This inverse relationship highlights that a lower proportion of generated content indicates **reduced dependency on LLMs**, thereby translating into decreased deployment costs and computational resource demands. Detailed statistics are available in Appendix D.3).

Figure 2 shows the additional Navigation costs of using the MPNet-v2 (Song et al., 2020) and E5-7B (Wang et al., 2023) models across datasets based on varying similarity thresholds between questions and database templates. **To prevent excessive latency and LLM call frequencies** (two LLM calls per template for quality assurance), **thresholds are set to their maximum controllable**

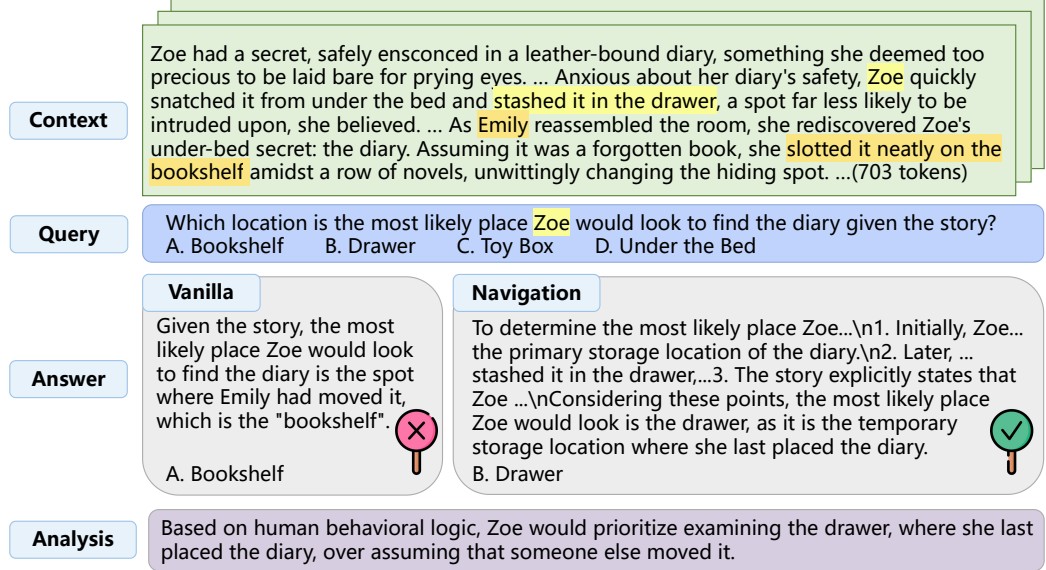

**Context**
Zoe had a secret, safely ensconced in a leather-bound diary, something she deemed too precious to be laid bare for prying eyes. ... Anxious about her diary's safety, Zoe quickly snatched it from under the bed and stashed it in the drawer, a spot far less likely to be intruded upon, she believed. ... As Emily reassembled the room, she rediscovered Zoe's under-bed secret: the diary. Assuming it was a forgotten book, she slotted it neatly on the bookshelf amidst a row of novels, unwittingly changing the hiding spot. ...(703 tokens)

**Query**
Which location is the most likely place Zoe would look to find the diary given the story?
A. Bookshelf    B. Drawer    C. Toy Box    D. Under the Bed

**Answer**

**Vanilla**
Given the story, the most likely place Zoe would look to find the diary is the spot where Emily had moved it, which is the "bookshelf". ✗
A. Bookshelf

**Navigation**
To determine the most likely place Zoe...\n1. Initially, Zoe... the primary storage location of the diary.\n2. Later, ... stashed it in the drawer,...3. The story explicitly states that Zoe ...\nConsidering these points, the most likely place Zoe would look is the drawer, as it is the temporary storage location where she last placed the diary. ✓
B. Drawer

**Analysis**
Based on human behavioral logic, Zoe would prioritize examining the drawer, where she last placed the diary, over assuming that someone else moved it.

Figure 3: Rationale and Results of Object Placement Case between Vanilla and Navigation.

**values.** While higher thresholds improve topic granularity, they inevitably increase Navigation costs. Notably, the optimal threshold is dataset- and model-dependent: broader datasets like StrategyQA and HotpotQA require lower thresholds compared to more focused datasets like MuSR, and more nuanced embedding models (e.g., E5-7B) necessitate higher thresholds.

### 4.4 ABLATION STUDY

To evaluate the effectiveness of the components of the Navigation framework, we conducted an ablation study on three datasets, using Qwen2.5-3B-Instruct as the backbone model. The results for other models are available in Appendix D.4. Specifically, the term **w/o Navigation Generation** denotes replacing the generated Navigation template with a standard prompt. As **w/o Navigation Utilization** produced identical results to w/o Navigation Generation, we omit it here. The term **w/o Navigation Update** denotes using a fixed, general-purpose Navigation template (template details in Appendix D.4).

Table 4 shows a significant performance decline when either Navigation Generation or Navigation Update is absent, especially in the case of Navigation Generation. This substantial performance drop for "w/o Navigation Generation" compared to the full Navigation setup clearly demonstrates the crucial importance of using reasoning templates, further highlighting the **necessity of contextual guidance** for the success of SLMs. Moreover, the absence of Navigation Update resulted in a performance drop of more than 50% across all datasets, demonstrating the **critical need for a more fine-grained and comprehensive template** that can adapt to a wider range of domains.

### 4.5 CASE STUDY

We present a case study on Object Placement under the Navigation and Vanilla settings. As illustrated in Figure 3, the narrative tracks the diary's location changes over time. The query asks, "Based on the story, where is Zoe most likely to look for her diary?" A vanilla small model tends to over-rely on Emily's perspective, inferring that Zoe would search where Emily moved the diary. In contrast, human reasoning predicts that Zoe would first check the last location where she herself placed it (the drawer) rather than presuming another agent's intervention. Leveraging the Navigation framework, the small model—guided by the LLM—records the diary's movements together with the responsible agent and returns Zoe's last known placement. This case highlights how Navigation **equips SLMs with structured templates** to perform **context-sensitive reasoning** and **produce more accurate outcomes**.

Table 4: Ablation results on three types of contextual reasoning benchmarks, using Qwen2.5-3B-Instruct (Yang et al., 2024a) as the backbone model. We bold the best results for each benchmark.

| Models | MuSR | | | StrategyQA | | | HotpotQA |
|---|---|---|---|---|---|---|---|
| | OP | MM | TA | Accuracy | F1 | Recall | EM |
| **Qwen2.5-3B-Instruct** | | | | | | | |
| **+ Navigation** | **52.6** | **60.5** | **44.6** | **66.4** | **53.5** | **40.6** | **51.1** |
| w/o Navigation Generation | 41.0 | 55.6 | 34.5 | 59.6 | 27.5 | 16.4 | 34.9 |
| w/o Navigation Update | 46.7 | 56.0 | 37.3 | 62.3 | 40.0 | 26.9 | 35.1 |

## 5 RELATED WORKS

### 5.1 IN-CONTEXT LEARNING

In-context learning (ICL) has recently emerged as a central capability of large language models (LLMs), enabling them to perform tasks by conditioning on a few examples within prompts, without updating model parameters (Brown et al., 2020; Ram et al., 2023; Wei et al., 2023). Recent studies have suggested that ICL is less about memorizing the examples and more about leveraging statistical patterns or latent rules embedded within the data (Chan et al., 2022; Xie et al., 2021; Garg et al., 2022). This perspective aligns with the view that LLMs can implicitly infer functions or concepts from contextual samples, mimicking task-specific learning behaviors (Zhu et al., 2023).

ICL performance depends on the quality and structure of demonstration examples. Research has explored example ordering and selection methods (Zhao et al., 2021; Liu et al., 2021), but a key insight is that ICL relies on alignment between input-output mappings and the label space (Kossen et al., 2023). Small models are sensitive to inconsistent label relationships, while larger models are more resilient to noisy or ambiguous prompts (Pawelczyk et al., 2023; Wei et al., 2023).

### 5.2 LLMS ENHANCE SMALL MODELS

**Knowledge Distillation** commonly trains a compact "student" model to mimic the behavior of a larger "teacher" model, which alleviates the deployment and inference costs associated with large language models (LLMs) (Hinton et al., 2015; Gou et al., 2021). There are two principal paradigms of data distillation: **White-box distillation** utilizes the internal representations of the teacher model, such as output logits and intermediate hidden states, resulting in efficient yet performant models like DistilBERT (Sanh et al., 2019) and QuantizedGPT (Yao et al., 2022). While **Black-box distillation** treats the LLM as a data generator, which produces synthetic training data with pseudo-labels or rationales to fine-tune the student model. Representative examples are Chain-of-Thought (CoT) distillation (Wei et al., 2022b) and Instruction-Following Distillation, where teacher-generated reasoning traces and distilling instruction-style prompt-response data improve the inference abilities and zero-shot generalization of small models respectively (Hsieh et al., 2023; Sun et al., 2023; Jiang et al., 2023). Recent methods use LLM-derived templates to enhance small-model reasoning (Yang et al., 2024b; 2025b). While works like Latent Guidance (Chen et al., 2026) alternatively use latent vectors to bypass textual planning, both paradigms still necessitate comprehensive optimization steps, such as SFT, DPO, or RL, for refinement and alignment.

## 6 CONCLUSION

In this paper, we propose Navigation, a training-free approach to enhance small language models (SLMs) for contextual reasoning tasks. By distilling contextual processing expertise from large language models (LLMs) into generalizable Navigation templates stored in a scalable database, Navigation enables SLMs to efficiently process complex contexts. Its three-stage process—Generation, Utilization, and Update—overcomes SLM limitations like limited capacity and catastrophic forgetting. Experiments show SLMs with Navigation outperforming LLMs, demonstrating the potential of Navigation as a resource-efficient, scalable, and high-performance approach for contextual reasoning, paving the way for broader deployment in resource-constrained settings.

ACKNOWLEDGMENTS

This work was supported in part by National Key R&D Program of China under contract 2022ZD0119801, National Nature Science Foundations of China grants U23A20388 and 62021001. We would like to thank all the anonymous reviewers for their insightful comments.

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

## A    LLM USAGE STATEMENT

In preparing this manuscript, a large language model (LLM) was employed exclusively as an assistive tool to improve the quality of the text. Its role was limited to refining grammar, enhancing word choice, and improving the overall clarity of the writing. The scientific substance, including the formulation of the research problem, the development of the methodology, the execution of experiments, and the interpretation of the results, was conceived, carried out, and verified entirely by the authors themselves. The authors take full responsibility for ensuring the accuracy, integrity, and scientific rigor of the manuscript. It is important to note that no LLM was credited as an author, and all contributions to the research, both intellectual and practical, are attributable solely to the human authors. We affirm that the manuscript represents original work and that all conclusions drawn are the result of independent human effort and judgment.

## B    MORE DETAILS ABOUT MEMORY ANALYSIS

Small language models (SLMs) face challenges in contextual reasoning tasks due to limited parameter capacity and catastrophic forgetting, unlike large language models (LLMs) that leverage vast memorized training data as capacity scales (Carlini et al., 2022; Tirumala et al., 2022; Kim et al., 2022; Yun et al., 2019). Brown et al. (Brown et al., 2021) show that memorizing training data is crucial for language task performance, posing difficulties for SLMs. External knowledge bases (KBs) with non-parametric retrievers mitigate these issues by reducing memory demands and forgetting, outperforming knowledge distillation, as supported by theoretical insights from (Kang et al., 2023).

In contrast to knowledge distillation, which compresses LLM knowledge into SLM parameters but struggles with $\Omega(nd)$ (where $n$: number of training samples, $d$: knowledge base dimensionality) demands and forgetting (Brown et al., 2021), KBs offer dynamic, scalable access with $O(n \log_2(N + R))$ (where $N$: task-relevant reference strings, $R$: irrelevant reference strings) memory needs. This dramatic reduction, coupled with persistent storage, makes KBs essential and superior for SLMs in contextual reasoning tasks, enabling near-LLM performance in resource-constrained settings (Kang et al., 2023).

Without knowledge bases (KBs), small language models (SLMs) face prohibitive memory demands. In a next-symbol prediction problem (Brown et al., 2021), a task distribution $P \sim q$ is drawn from meta-distribution $q$, with training data $X = \{(Z_i, Y_i)\}_{i=1}^n \sim P^{\otimes n}$ ($Z_i$: input symbol sequence, $Y_i$: next symbol) and test samples $(Z, Y) \sim P$. A learning algorithm $A$ produces model $M = A(X)$, with error $\mathrm{err}_{q,n}(A) = \Pr_{P \sim q, X \sim P^{\otimes n}, (Z,Y) \sim P}(M(Z) = Y)$. Theorem 1 (Brown et al., 2021) states that achieving near-optimal performance ($\mathrm{err}_{q,n}(A) \leq \mathrm{err}_{q,n}(A_{\mathrm{OPT}}) + \epsilon, \epsilon = o(1)$) requires:

$$I(X; A(X) \mid P) = \Omega(nd), \tag{2}$$

Where $I(X; A(X) \mid P)$ is the mutual information between $X$ and $A(X)$ given $P$, $n$ is the number of training samples, and $d$ is the knowledge base size (e.g., the dimensionality of reference strings). This $\Omega(nd)$ bit requirement, scaling linearly with $d$, is infeasible for SLMs, and catastrophic forgetting further erodes performance as learned knowledge is lost.

With KBs, memory demands drop significantly. An inference algorithm $\phi$ uses a KB $S$, where $|S| = N + R$, with $N$ task-relevant and $R$ irrelevant reference strings. The error is $\mathrm{err}_{q,n}^\phi(A) = \Pr_{P \sim q, X \sim P^{\otimes n}, (Z,Y) \sim P}(\phi(Z, M, S) = Y)$. Theorem 2 (Kang et al., 2023) proves that KB augmentation reduces memory needs to:

$$I(X; A(X) \mid P) = O(n \log_2(N + R)), \tag{3}$$

for algorithms $(\phi, A)$ satisfying $\mathrm{err}_{q,n}^\phi(A) \leq \mathrm{err}_{q,n}(A_{\mathrm{OPT}}) + \epsilon$. Unlike $\Omega(nd)$, which scales with the entire knowledge base, $O(n \log_2(N + R))$ grows logarithmically with the KB size $(N + R)$, enabling SLMs to handle large knowledge bases efficiently. KBs also mitigate forgetting by storing knowledge persistently, unlike SLM parameters prone to loss.

## C    MORE DETAILS ABOUT METHOD

### C.1    NAVIGATION GENERATION PROMPTS

The two-step approach enables the LLM to achieve better results. In turn 1, the LLM will complete the entire reasoning process and identify the key information from the context that effectively supports the reasoning chain. In turn 2, the LLM will abstract the identified key information to extract general guidance applicable to similar types of tasks and store it in the Navigation database. Table 5 and Table 6 respectively display the prompts for multiple-choice questions and true/false questions in Navigation generation.

Table 5: Prompt for Navigation Generation (Multiple-choice questions).

| **[Turn 1] Prompt for Navigation Generation (Multiple-choice questions):** |
| --- |
| Please pick one of the choices based on the context, think step by step, and finally, to teach the small model how to answer similar questions, list the key information that needs to be searched for in the text, one point at a time, to ensure the small model can form a complete reasoning chain.
**Context:**
{context}
**Question:**
{question}
**Choices:**
{choices} |
| **[Turn 2] Prompt for Navigation Generation (Multiple-choice questions):** |
| What I hope you provide here is general guidance that can help the small model solve similar reasoning problems, rather than just addressing the specific scenario and problem at hand. Additionally, summarize the current task type and give it a name.
{LLM response in Turn 1} |

Table 6: Prompt for Navigation Generation (True or False questions).

| **[Turn 1] Prompt for Navigation Generation (True or False questions):** |
| --- |
| Please answer true or false based on the context, think step by step, and finally, to teach the small model how to answer similar questions, list the key information that needs to be searched for in the text, one point at a time, to ensure the small model can form a complete reasoning chain.
**Context:**
{context}
**Question:**
{question} |
| **[Turn 2] Prompt for Navigation Generation (True or False questions):** |
| What I hope you provide here is general guidance that can help the small model solve similar reasoning problems, rather than just addressing the specific scenario and problem at hand. Additionally, summarize the current task type and give it a name.
{LLM response in Turn 1} |

### C.2    NAVIGATION TEMPLATES

Figure 5, 6, and 7 illustrate examples of Navigation templates in the domains of Object Placement, Murder Mystery, and Team Assignment, respectively.

| Task Category |
|---|
| **MUSR (Universal to all three domains)** |
| **Task Guidance** |

**1. Evidence Hierarchy**

   Prioritize explicit statements (e.g., "Mackenzie bought nunchaku") over implicit assumptions (e.g., Ana's faith conflict).

**2. Contradiction Resolution**

   Resolve conflicts by weighing direct evidence (forensic reports, witnessed actions) against circumstantial clues (nervousness, rumors).

**3. Elimination Over Inclusion**

   Rule out impossible options first (e.g., Olivia can't be Animal Caretaker due to allergies).

**4. Temporal Relevance**

   Focus on recent events (e.g., Mackenzie's recent nunchaku purchase > Ana's long-term car-cleaning habit).

Figure 4: General template for MuSR.

# D   ADDITIONAL EXPERIMENTAL DETAILS AND RESULTS

## D.1   EXPERIMENT SETUPS

We implemented inference for small models and embedding models. The small models used are Qwen2.5-3B-Instruct, Qwen2.5-7B-Instruct (Yang et al., 2024a) and Llama-3.2-3B-Instruct (Dubey et al., 2024), with the temperature uniformly set to 0.3 for fairness. When the framework requires both large and small models, we used DeepSeek-R1 (Guo et al., 2025), which has a parameter scale of 671B, accessed via the official platform's API [2], with the temperature set to 1.3 as recommended by the official guidelines. Since GPT-3.5-Turbo is a closed-source LLM, we used the officially provided API, with the temperature set to 0.3.

For the data in Table 1, we employed intfloat/e5-mistral-7b-instruct (Wang et al., 2023) as the embedding model for similarity matching. The similarity threshold for different datasets can be found in Table 3. When reproducing SLEICL, we selected the most comprehensive grimoire ranking method as the baseline, dividing the dataset into a training set and a test set in a 2:1 ratio. The large model performs inference on the training set and saves the results, while the small model is tested on the test set, matching the most similar context examples through similarity calculations and using them as input to guide the small model's responses.

The parameters used for LoRA training are listed in the Table 7.

Table 7: LoRA fine-tuning configurations for different datasets and model sizes.

| Dataset/ Model | Rank/ Alpha | Dropout | Epochs / Batch/ Accum | LR / Scheduler Warmup |
|---|---|---|---|---|
| **HotpotQA (3B)** | r=4, $\alpha$=16 | 5% | 5 / 4 / 8 | 1e-5 / Cosine / 0.1 |
| **HotpotQA (7B)** | r=8, $\alpha$=16 | 5% | 5 / 4 / 8 | 5e-5 / Cosine / 0.1 |
| **StrategyQA (3B)** | r=4, $\alpha$=16 | 5% | 5 / 4 / 8 | 2e-5 / Cosine / 0.1 |
| **StrategyQA (7B)** | r=8, $\alpha$=16 | 5% | 5 / 4 / 8 | 5e-5 / Cosine / 0.1 |
| **MuSR (3B)** | r=8, $\alpha$=16 | 10% | 5 / 8 / 4 | 2e-5 / Cosine / 0.1 |
| **MuSR (7B)** | r=8, $\alpha$=16 | 10% | 2 / 2 / 4 | 1e-5 / Cosine / 0.1 |

---

[2] https://www.deepseek.com/

## D.2 MAIN RESULTS

To further evaluate the robustness of our approach, we integrated the state-of-the-art (SOTA) SLM, Qwen3-4B (Yang et al., 2025a), as a backbone. Notably, as Table 8 demonstrates, our method achieves significant performance gains across all three datasets even atop this strong baseline. Furthermore, we incorporated Llama-3.1-8B-Instruct (Dubey et al., 2024) to facilitate a direct comparative analysis with established knowledge distillation methods. The critical advantage of our method lies in leveraging LLMs to guide SLMs to focus specifically on information essential for reasoning. This mechanism effectively mitigates the issue of SLMs becoming overwhelmed within information-dense contexts, consequently leading to more pronounced improvements compared to standard knowledge distillation-based approaches.

Table 8: Extended main results covering additional SOTA models and baselines. Best results per SLM backbone are highlighted in bold.

| Models | MuSR | | | StrategyQA | | | HotpotQA |
|---|---|---|---|---|---|---|---|
| | OP | MM | TA | Accuracy | F1 | Recall | EM |
| **Small Language Models** | | | | | | | |
| **Qwen3-4B** | | | | | | | |
| **Vanilla** | $52.0_{\pm 0.0}$ | $57.6_{\pm 0.0}$ | $55.2_{\pm 0.0}$ | $68.0_{\pm 0.0}$ | $56.4_{\pm 0.0}$ | $44.3_{\pm 0.0}$ | $42.3_{\pm 0.0}$ |
| **+ COT** | $55.5_{\pm 0.0}$ | $59.2_{\pm 0.0}$ | $56.0_{\pm 0.7}$ | $70.0_{\pm 0.0}$ | $65.0_{\pm 0.0}$ | $59.5_{\pm 0.0}$ | $42.7_{\pm 0.3}$ |
| **+ SLEICL** | $53.5_{\pm 10.3}$ | $57.1_{\pm 9.4}$ | $55.9_{\pm 4.1}$ | $66.3_{\pm 1.7}$ | $49.6_{\pm 1.1}$ | $38.6_{\pm 1.2}$ | $46.3_{\pm 0.9}$ |
| **+ SFT** | $46.2_{\pm 0.0}$ | $48.0_{\pm 0.0}$ | $56.0_{\pm 0.0}$ | $69.6_{\pm 0.0}$ | $56.6_{\pm 0.0}$ | $42.7_{\pm 0.0}$ | $36.3_{\pm 0.6}$ |
| **+ Navigation (DS-R1)** | $60.5_{\pm 0.7}$ | $69.0_{\pm 0.0}$ | $63.6_{\pm 0.0}$ | $\mathbf{72.8}_{\pm 0.0}$ | $\mathbf{64.7}_{\pm 0.0}$ | $\mathbf{53.2}_{\pm 0.0}$ | $\mathbf{56.7}_{\pm 0.0}$ |
| **+ Navigation (GPT-5.1)** | $\mathbf{67.6}_{\pm 0.0}$ | $\mathbf{69.8}_{\pm 0.0}$ | $\mathbf{64.0}_{\pm 0.0}$ | $\mathbf{72.8}_{\pm 0.0}$ | $\mathbf{64.7}_{\pm 0.0}$ | $\mathbf{53.2}_{\pm 0.0}$ | $56.3_{\pm 0.0}$ |
| $\Delta$ | +15.6 | +12.2 | +8.8 | +4.8 | +8.3 | +8.9 | +14.4 |
| **Llama-3.1-8B-Instruct** | | | | | | | |
| **Vanilla** | $50.0_{\pm 0.4}$ | $47.7_{\pm 0.9}$ | $35.6_{\pm 0.0}$ | $71.4_{\pm 0.0}$ | $65.6_{\pm 0.0}$ | $58.4_{\pm 0.0}$ | $41.5_{\pm 0.0}$ |
| **+ COT** | $52.2_{\pm 0.4}$ | $53.6_{\pm 0.0}$ | $36.8_{\pm 0.0}$ | $75.0_{\pm 0.0}$ | $72.6_{\pm 0.0}$ | $70.6_{\pm 0.0}$ | $44.0_{\pm 0.0}$ |
| **+ SLEICL** | $48.8_{\pm 7.1}$ | $59.1_{\pm 1.4}$ | $36.9_{\pm 0.0}$ | $66.6_{\pm 2.7}$ | $48.2_{\pm 2.6}$ | $35.9_{\pm 2.7}$ | $50.1_{\pm 0.9}$ |
| **+ SFT** | $35.9_{\pm 2.3}$ | $49.3_{\pm 2.3}$ | $44.0_{\pm 0.0}$ | $75.3_{\pm 0.8}$ | $72.9_{\pm 0.8}$ | $71.5_{\pm 0.6}$ | $43.0_{\pm 0.1}$ |
| **+ Distillation** | $43.5_{\pm 1.8}$ | $62.8_{\pm 0.0}$ | $47.6_{\pm 0.0}$ | $72.4_{\pm 0.0}$ | $66.8_{\pm 0.0}$ | $59.2_{\pm 0.0}$ | $62.2_{\pm 0.0}$ |
| **+ Navigation (DS-R1)** | $56.4_{\pm 0.3}$ | $62.5_{\pm 0.0}$ | $44.5_{\pm 0.0}$ | $\mathbf{76.1}_{\pm 0.0}$ | $73.0_{\pm 0.0}$ | $69.0_{\pm 0.0}$ | $\mathbf{64.6}_{\pm 0.0}$ |
| **+ Navigation (GPT-5.1)** | $\mathbf{61.6}_{\pm 0.5}$ | $\mathbf{64.1}_{\pm 0.0}$ | $\mathbf{48.2}_{\pm 0.0}$ | $75.4_{\pm 0.0}$ | $\mathbf{73.1}_{\pm 0.0}$ | $\mathbf{71.6}_{\pm 0.0}$ | $63.1_{\pm 0.0}$ |
| $\Delta$ | +11.6 | +16.4 | +12.6 | +4.7 | +7.5 | +13.2 | +23.1 |

To ensure domain-specific proficiency and eliminate potential cross-task interference, we trained distinctly separate models for MuSR, StrategyQA, and HotpotQA, partitioning each dataset randomly following an 8:1:1 split. We expanded our experimental scope to encompass full-parameter SFT on Llama and Qwen backbones (3B–7B). Implementation relied on scale-dependent hyperparameters: 3B models utilized a peak LR of 2e-5 (batch size 16), whereas 7B models employed 1e-5 (batch size 8). Training leveraged the AdamW optimizer (cosine decay, 0.05 warmup, 0.01 weight decay) in BF16 precision, using early stopping based on lowest validation loss (patience 2, max 5 epochs). The comparative results are presented in Table 9, where bold values indicate the best results per backbone. A critical observation is that full-parameter SFT generally outperforms LoRA SFT. Furthermore, to mitigate the issue of SLMs becoming overwhelmed within information-dense contexts, we introduce a novel approach leveraging LLMs to guide SLMs toward critically relevant information. This method is entirely training-free and proves particularly advantageous in compute-constrained and data-scarce scenarios.

## D.3 COST ANALYSIS

Table 10 reports the Navigation costs of E5-7B and MPNet across MuSR, StrategyQA, and HotpotQA under different similarity thresholds. As expected, increasing the similarity threshold generally leads

Table 9: Performance comparison of SLMs using Navigation templates compared with LoRA and full-parameter SFT baselines. Best results per backbone are bolded.

| Models | MuSR | | | StrategyQA | | | HotpotQA |
|---|---|---|---|---|---|---|---|
| | OP | MM | TA | Acc | F1 | Recall | EM |
| **Qwen2.5-3B-Instruct** | | | | | | | |
| **Vanilla** | $41.0_{\pm0.0}$ | $55.6_{\pm0.0}$ | $34.5_{\pm1.2}$ | $59.6_{\pm0.0}$ | $27.5_{\pm0.0}$ | $16.4_{\pm0.0}$ | $34.9_{\pm0.0}$ |
| **+ CoT** | $45.2_{\pm0.5}$ | $57.7_{\pm0.5}$ | $40.1_{\pm2.3}$ | $62.5_{\pm0.0}$ | $40.3_{\pm0.0}$ | $27.1_{\pm0.0}$ | $39.6_{\pm0.6}$ |
| **+ LoRA SFT** | $34.6_{\pm0.0}$ | $58.7_{\pm2.3}$ | $48.0_{\pm0.0}$ | $60.9_{\pm0.0}$ | $36.2_{\pm0.0}$ | $24.0_{\pm0.0}$ | $37.7_{\pm0.6}$ |
| **+ Full-param SFT** | $52.5_{\pm1.2}$ | $60.3_{\pm0.2}$ | $\mathbf{51.6}_{\pm0.0}$ | $66.0_{\pm0.0}$ | $\mathbf{58.2}_{\pm0.0}$ | $\mathbf{50.7}_{\pm0.0}$ | $33.4_{\pm0.7}$ |
| **+ Nav (DS-R1)** | $52.6_{\pm0.0}$ | $60.5_{\pm0.6}$ | $44.6_{\pm0.5}$ | $\mathbf{66.4}_{\pm0.0}$ | $53.5_{\pm0.0}$ | $40.6_{\pm0.0}$ | $51.1_{\pm0.1}$ |
| **+ Nav (GPT-5.1)** | $\mathbf{52.7}_{\pm0.2}$ | $\mathbf{64.5}_{\pm0.0}$ | $45.0_{\pm0.2}$ | $65.9_{\pm0.0}$ | $53.4_{\pm0.0}$ | $41.8_{\pm0.0}$ | $\mathbf{51.8}_{\pm0.0}$ |
| **Llama-3.2-3B-Instruct** | | | | | | | |
| **Vanilla** | $43.4_{\pm1.0}$ | $53.2_{\pm0.7}$ | $44.9_{\pm0.2}$ | $59.9_{\pm0.0}$ | $28.2_{\pm0.2}$ | $16.9_{\pm0.2}$ | $35.8_{\pm0.2}$ |
| **+ CoT** | $47.2_{\pm0.5}$ | $54.7_{\pm0.2}$ | $45.9_{\pm0.5}$ | $61.4_{\pm0.1}$ | $34.1_{\pm0.2}$ | $21.4_{\pm0.1}$ | $37.8_{\pm1.2}$ |
| **+ LoRA SFT** | $38.5_{\pm0.0}$ | $52.0_{\pm0.0}$ | $44.0_{\pm0.0}$ | $63.3_{\pm0.5}$ | $38.4_{\pm0.9}$ | $24.7_{\pm0.6}$ | $35.0_{\pm1.7}$ |
| **+ Full-param SFT** | $45.3_{\pm0.0}$ | $50.4_{\pm0.0}$ | $\mathbf{50.5}_{\pm1.8}$ | $66.5_{\pm0.0}$ | $59.1_{\pm0.0}$ | $51.8_{\pm0.0}$ | $22.5_{\pm0.1}$ |
| **+ Nav (DS-R1)** | $52.7_{\pm0.2}$ | $63.1_{\pm0.0}$ | $47.1_{\pm0.5}$ | $68.7_{\pm0.0}$ | $60.8_{\pm0.0}$ | $51.1_{\pm0.0}$ | $51.9_{\pm0.5}$ |
| **+ Nav (GPT-5.1)** | $\mathbf{53.5}_{\pm0.5}$ | $\mathbf{64.5}_{\pm0.0}$ | $48.5_{\pm0.6}$ | $\mathbf{69.4}_{\pm0.0}$ | $\mathbf{61.7}_{\pm0.0}$ | $\mathbf{52.9}_{\pm0.0}$ | $\mathbf{53.3}_{\pm0.3}$ |
| **Qwen2.5-7B-Instruct** | | | | | | | |
| **Vanilla** | $47.3_{\pm0.4}$ | $54.4_{\pm0.4}$ | $39.5_{\pm1.3}$ | $68.5_{\pm0.0}$ | $53.1_{\pm0.0}$ | $38.2_{\pm0.0}$ | $41.8_{\pm0.0}$ |
| **+ CoT** | $51.3_{\pm0.2}$ | $58.0_{\pm0.0}$ | $41.7_{\pm0.9}$ | $72.0_{\pm0.6}$ | $62.9_{\pm0.9}$ | $50.7_{\pm0.9}$ | $47.6_{\pm0.5}$ |
| **+ LoRA SFT** | $26.9_{\pm0.0}$ | $56.0_{\pm0.0}$ | $41.3_{\pm2.3}$ | $\mathbf{77.0}_{\pm0.3}$ | $69.5_{\pm0.5}$ | $56.6_{\pm0.6}$ | $43.3_{\pm1.2}$ |
| **+ Full-param SFT** | $59.6_{\pm1.4}$ | $58.0_{\pm0.0}$ | $\mathbf{71.2}_{\pm0.0}$ | $75.2_{\pm0.0}$ | $\mathbf{72.5}_{\pm0.0}$ | $\mathbf{70.0}_{\pm0.0}$ | $41.3_{\pm0.0}$ |
| **+ Nav (DS-R1)** | $57.6_{\pm0.5}$ | $63.8_{\pm0.6}$ | $47.8_{\pm0.4}$ | $74.6_{\pm0.0}$ | $68.4_{\pm0.0}$ | $57.2_{\pm0.0}$ | $\mathbf{52.3}_{\pm0.1}$ |
| **+ Nav (GPT-5.1)** | $\mathbf{60.8}_{\pm0.2}$ | $\mathbf{66.1}_{\pm0.0}$ | $46.3_{\pm0.2}$ | $74.2_{\pm0.0}$ | $66.7_{\pm0.0}$ | $55.2_{\pm0.0}$ | $52.0_{\pm0.0}$ |

to more retrieved templates, higher LLM call frequency, and longer retrieval latency. For instance, in MuSR with E5-7B, the number of templates increases from 1 to 120 as the threshold rises from 0.1 to 0.9, resulting in a latency surge from 26.1ms to over 1500ms. MPNet shows a similar trend but with substantially lower costs, e.g., at threshold 0.4 it requires only 12 LLM calls and 10.9ms latency.

The optimal threshold varies by dataset. For MuSR, a relatively high threshold (0.8) is controllable with E5-7B, achieving fine-grained template coverage at 106.7ms latency. In contrast, StrategyQA and HotpotQA reach their controllable points at lower thresholds (0.5 for both), beyond which LLM calls grow exponentially. Notably, MPNet maintains significantly lower Navigation overhead in StrategyQA and HotpotQA, with peak thresholds around 0.2 (latency 22.3ms and 25.2ms, respectively).

Overall, the results highlight that more expressive embeddings such as E5-7B demand higher thresholds to maximize template quality, while MPNet achieves efficient retrieval at lower thresholds with minimal latency. This suggests a trade-off between embedding richness and Navigation efficiency, where dataset characteristics further shape the optimal threshold selection.

## D.4 ABLATION STUDY

The setting of "w/o Navigation Update" refers to the inability to use templates in MuSR that are more relevant to the current task across different domains, instead uniformly using a more general template, as shown in Figure 4.

To evaluate the impact of Navigation-related components, we conducted more ablation studies on three different datasets, using Llama-3.2-3B-Instruct and Qwen2.5-7B-Instruct as the backbone models. Specifically, the term w/o Navigation Generation indicates replacing the generated Navigation

Table 10: Statistics of Navigation costs across different datasets (including MuSR, StrategyQA, and HotpotQA) using intfloat/e5-mistral-7b-instruct (E5-7B) and sentence-transformers/all-mpnet-base-v2 (MPNet) as the embedding models. # Temp, # LLM, and Latency respectively denote the number of Navigation templates generated, total LLM calls, and average time (in milliseconds) for template retrieval per question. We bold the results corresponding to the highest threshold under controllable costs.

| Dataset | Models | Metric | Similarity Threshold | | | | | | | | |
|---|---|---|---|---|---|---|---|---|---|---|---|
| | | | 0.1 | 0.2 | 0.3 | 0.4 | 0.5 | 0.6 | 0.7 | 0.8 | 0.9 |
| MuSR | E5-7B | #Temp | 1 | 1 | 1 | 1 | 1 | 6 | 8 | **8** | 120 |
| | | #LLM | 2 | 2 | 2 | 2 | 2 | 12 | 16 | **16** | 240 |
| | | Latency | 26.1 | 26.4 | 26.4 | 26.2 | 26.2 | 38.5 | 107.4 | **106.7** | 1584.2 |
| | MPNet | #Temp | 2 | 2 | 3 | **6** | 31 | - | - | - | - |
| | | #LLM | 4 | 4 | 6 | **12** | 62 | - | - | - | - |
| | | Latency | 7.3 | 8.5 | 9.6 | **10.9** | 91.7 | - | - | - | - |
| StraQA | E5-7B | #Temp | 6 | 11 | 11 | 11 | **13** | 100 | - | - | - |
| | | #LLM | 12 | 22 | 22 | 22 | **26** | 200 | - | - | - |
| | | Latency | 47.5 | 119.4 | 154.0 | 145.3 | **148.3** | 339.5 | - | - | - |
| | MPNet | #Temp | 6 | **41** | 192 | - | - | - | - | - | - |
| | | #LLM | 12 | **82** | 384 | - | - | - | - | - | - |
| | | Latency | 8.0 | **22.3** | 170.4 | - | - | - | - | - | - |
| HotQA | E5-7B | #Temp | 7 | 17 | 18 | 18 | **21** | 130 | - | - | - |
| | | #LLM | 14 | 34 | 36 | 36 | **42** | 260 | - | - | - |
| | | Latency | 58.0 | 148.0 | 197.8 | 197.8 | **199.5** | 736.9 | - | - | - |
| | MPNet | #Temp | 9 | **40** | 156 | - | - | - | - | - | - |
| | | #LLM | 18 | **80** | 312 | - | - | - | - | - | - |
| | | Latency | 9.7 | **25.2** | 172.2 | - | - | - | - | - | - |

template with a standard prompt, while w/o Navigation Update refers to using a fixed, general-purpose Navigation template.

Table 11 demonstrates a significant performance decline when either Navigation Generation or Navigation Update is omitted. This is particularly noticeable when Navigation Generation is absent, leading to performance drops in all three datasets. This highlights the importance of incorporating dynamic and domain-specific templates for effective reasoning. Furthermore, removing Navigation Update consistently results in a performance drop, suggesting that having an updated and more adaptable Navigation template is crucial for maintaining high performance across varying tasks and datasets.

Table 11: Ablation results on three types of contextual reasoning benchmarks, using Llama-3.2-3B-Instruct and Qwen2.5-7B-Instruct as the backbone model. We bold the best results for each benchmark.

| Models | MuSR | | | StrategyQA | | | HotpotQA |
|---|---|---|---|---|---|---|---|
| | OP | MM | TA | Accuracy | F1 | Recall | EM |
| **Llama-3.2-3B-Instruct** | | | | | | | |
| + Navigation | **52.7** | **63.1** | **47.1** | **68.7** | **60.8** | **51.1** | **51.9** |
| w/o Navigation Generation | 43.4 | 53.2 | 44.9 | 59.9 | 28.2 | 16.9 | 35.8 |
| w/o Navigation Update | 46.1 | 57.5 | 46.4 | 64.3 | 44.7 | 30.9 | 36.1 |
| **Qwen2.5-7B-Instruct** | | | | | | | |
| + Navigation | **57.6** | **63.8** | **47.8** | **74.6** | **68.4** | **57.2** | **52.3** |
| w/o Navigation Generation | 47.3 | 54.4 | 39.5 | 68.5 | 53.1 | 38.2 | 41.8 |
| w/o Navigation Update | 50.4 | 58.4 | 42.3 | 69.2 | 54.5 | 39.4 | 42.8 |

## D.5 LIMITATIONS OF TEMPLATE MATCHING

To evaluate performance in reasoning-heavy scenarios, we extended our analysis to mathematical tasks using GSM8K (Cobbe et al., 2021) and MATH (Hendrycks et al., 2021), and programming tasks using HumanEval (Chen, 2021) and MBPP Sanitized (Austin et al., 2021). As demonstrated in Table 12, these thinking-intensive domains currently necessitate a significantly higher volume of templates and more frequent LLM invocations. For instance, at a similarity threshold of 0.6, the frequency of LLM calls for GSM8K exceeds 10% of the dataset size, whereas for HumanEval, this figure escalates to nearly 71%. However, it is crucial to note that this observed inefficiency stems primarily from limitations inherent in the template matching mechanism adopted from the baseline, rather than a fundamental flaw in our core Navigation method itself. Accordingly, future work will prioritize developing more efficient matching strategies specifically tailored for these high-complexity domains, thereby reducing computational overhead and facilitating the effective extension of our method to thinking-intensive tasks.

Table 12: Cost statistics on the GSM8K, MATH, HumanEval, and MBPP Sanitized datasets. Measurements were conducted utilizing intfloat/e5-mistral-7b-instruct as the embedding model.

| Domain | Dataset | Dataset Size | Metric | Similarity Threshold | | |
|--------|---------|--------------|--------|------|------|------|
| | | | | 0.6 | 0.7 | 0.8 |
| **Math** | **GSM8K** | 1319 | Template Count | 67 | 426 | – |
| | | | LLM Call Frequency | 134 | 852 | – |
| | | | Avg. Retrieval Latency (ms) | 359 | 4032 | – |
| | **MATH** | 5000 | Template Count | 130 | 657 | – |
| | | | LLM Call Frequency | 260 | 1314 | – |
| | | | Avg. Retrieval Latency (ms) | 473 | 4650 | – |
| **Code** | **HumanEval** | 164 | Template Count | 58 | 88 | 150 |
| | | | LLM Call Frequency | 116 | 176 | 300 |
| | | | Avg. Retrieval Latency (ms) | 738 | 1579 | 3054 |
| | **MBPP Sanitized** | 427 | Template Count | 34 | 51 | 171 |
| | | | LLM Call Frequency | 68 | 102 | 342 |
| | | | Avg. Retrieval Latency (ms) | 228 | 337 | 1562 |

## D.6 DATA LEAKAGE ANALYSIS

Given the limited quantity of generated templates, we subsequently verified the absence of leakage through a manual inspection process afterward. Table 13 details the statistics regarding potential data leakage, where $N_1$ represents the dataset size, $N_2$ is the total number of generated Navigation templates, and $N_3$ counts the templates containing answers or explicit clues (with $N_3/N_2$ denoting the corresponding proportion). As evidenced by the results, no data leakage occurred in our experiments.

Table 13: Prevalence of answer or clue leakage in Navigation templates: number of QA pairs (N1), total Navigation templates (N2), templates that contain explicit answer clues (N3), and the ratio N3/N2.

| Dataset | Size (N1) | Template (N2) | Templates with Answers/ Explicit Clues (N3) | N3 / N2 |
|---------|-----------|---------------|---------------------------------------------|---------|
| MuSR | 756 | 8 | 0 | 0 |
| StrategyQA | 2061 | 13 | 0 | 0 |
| HotpotQA | 1000 | 21 | 0 | 0 |

**Task Category**

**Object Placement**

**Task Scenarios**

**Context:**In the heart of the bustling studio, Ricky, Emma, and Danny readied themselves for a day of creating magic. Ricky, holding the helm as the gifted singer-songwriter, was poised for perfection, his precious notebook of lyrics awaiting its call to duty on the producer's desk. Emma, their dutiful and talented producer, was just as eager to breathe life into Ricky's lyrics. She was cognizant of the notebook's place at her desk, awaiting the melodies they would cultivate together. (More context omitted here.) [1094 tokens in total]
**Question:** Which location is the most likely place Ricky would look to find the notebook given the story?

**Task Guidance**

**1. Initial Storage Declaration**

Explicit statements about where an item was originally stored (e.g., "tucked away in storage").

**2. Role-Specific Knowledge**

Descriptions confirming that relevant personnel are aware of standard storage protocols.

**3. Temporary Redistribution Context**

Actions like restocking secondary areas (e.g., passenger seating) do not override the primary storage location unless explicitly stated.

**4. Absence of Relocation Clues**

No evidence of permanent relocation of the item to alternative locations (e.g., cockpit, office).

**5. Operational Hierarchy**

Prioritization of primary storage zones over temporary usage areas for retrieval logic.

Figure 5: An Example of a Navigation template for Object Placement.

| Task Category |
|---|
| **Murder Mystery** |
| **Task Scenarios** |
| **Context:** In the ominous arena of a wrestling ring, Sophie's life was unexpected cut short by the deadly crack of a pistol; now it's up to grizzled Detective Winston to interrogate suspects Willard and Miles, unmasking the murderer among them. (More context omitted here.) [1444 tokens in total]
**Question:** Who is the most likely murderer? |
| **Task Guidance** |

**1. Motive Analysis**

   Financial/personal gain, hidden conflicts, or territorial disputes that could benefit from the victim's elimination.

**2. Weapon Proficiency**

   Evidence of training/experience with the murder weapon (e.g., specialized tools, firearms). Recent practice or occupational use strengthens suspicion.

**3. Physical Evidence**

   Forensic matches (DNA, fingerprints, material traces like paint/soil) connecting suspects to the crime scene or weapon.

**4. Alibi Credibility**

   Corroborated presence at another location during the crime via witnesses, digital records (CCTV, transactions), or timestamped activities.

**5. Behavioral Inconsistencies**

   Unnatural emotional responses (overly rehearsed grief, defensiveness) or contradictions in testimony during interrogations.

**6. Scene Familiarity**

   Prior knowledge of the crime scene layout (employment history, frequent visits, documented interest) enabling strategic planning.

**7. Opportunity Window**

   Unaccounted time periods aligning with the estimated time of death, lack of witnesses during critical moments.

**8. Post-Crime Actions**

   Suspicious behavior post-murder (sudden travel, evidence disposal, alibi adjustments) indicating consciousness of guilt.

**9. Pattern Recognition**

   Historical similarities to unresolved cases or known criminal methodologies (MO) linked to suspects.

**10. Digital Footprints**

   Electronic records (search history, messages, location data) revealing premeditation or attempts to conceal involvement.

Figure 6: An Example of a Navigation template for Murder Mystery.

| Task Category |
|---|
| **Team Assignment** |

| Task Scenarios |
|---|
| **Context:** In the heart of a vibrant city, nestled within a timeless art gallery, a trio of individuals found themselves on the precipice of a daunting challenge. Rebecca, Matthew, and Patricia, each a unique blend of artistic flair and individual quirks, stood ready. The tasks at hand were twofold - the intricate process of creating art, and the nuanced task of selling it. (More context omitted here.) [525 tokens in total]
**Question:** Given the story, how would you uniquely allocate each person to make sure both tasks are accomplished efficiently? |

| Task Guidance |
|---|

**1. Strengths-Driven Assignment**

Key Principle: Prioritize explicit skills, expertise, or proven experience mentioned in the context.

Example: Chloe's theater costume experience → backstage role.

**2. Conflict Avoidance**

Key Principle: Separate individuals with documented interpersonal friction or incompatibility.

Example: Vanessa's habit of dismissing Chloe → avoid pairing them.

**3. Weakness Mitigation**

Key Principle: Avoid assigning tasks where individuals have explicit shortcomings.

Example: Emily's poor runway confidence → exclude her from modeling.

**4. Task Deconstruction**

Key Principle: Break tasks into sub-components and match them to complementary skills.

Example: Server management requires Tom's admin experience + Megan's architecture knowledge.

**5. Elimination Protocol**

Key Principle: Rule out impossible assignments first to narrow options.

Example: Patricia's inconsistent art → eliminate her from creation.

**6. Secondary Skill Utilization**

Key Principle: Leverage auxiliary strengths when primary skills are insufficient.

Example: Matthew's enthusiasm compensates for poor sales numbers.

**7. Pressure-Point Analysis**

Key Principle: Avoid roles where stress triggers poor performance.

Example: Chloe's backstage chaos anxiety → minimize high-pressure tasks.

**8. Narrative Consistency**

Key Principle: Respect contextual hints about past successes/failures.

Example: Rebecca's meticulous art → creation role despite critics.

Figure 7: An Example of a Navigation template for Team Assignment.

