# OpenReview forum: "Following the Navigation: Enhancing Small Language Models Contextual Reasoning with LLM Guidance"
_ICLR.cc/2026/Conference — ICLR 2026 Poster_

### Official Review · Reviewer_Gxds · 2025-10-15

**Soundness:** 2
**Presentation:** 2
**Contribution:** 2
**Rating:** 4
**Confidence:** 4

**Summary:**

This paper presents an approach called Navigation to distill LLMs for SLMs for better performance without additional training. The method can be decomposed into three stages, i.e., Generation, Utilization, and Update, and yield better performance.

**Strengths:**

- Navigation serves as a good simulator for using knowledge from LLMs to help SLMs boost performance.
- The authors provide detailed experimental results on four SLMs to prove the effectiveness.

**Weaknesses:**

- For the method, the authors lack some important details. 1. During Guidance Generalization, is the same test set where SLMs are evaluated on used? Is it only the questions correctly answered by LLMs used, or are all the questions used for guidance generalization? Is the guidance consisting of only the template or the relevant reasoning part? Now it seems that the LLMs have everything for the given question, and the SLMs should just follow. Also, there is some potential for data leakage where LLMs may provide the final answers. 2. For Navigation Utilization, is it a multi-turn process where the SLMs first fill in the template, then get the answer, or is it an e2e process? 3. For Navigation Update, how do you ensure that no answer or clues are included in the template? Did you do some experiments or check some cases?
- For the experiments, why do SFT-models achieve lower performance than CoT-only methods? More implementation details should be included. Since the authors use LoRA for Fine-tuning, full SFT should also be included for fairer comparison. Table 7 presents that as the threshold of similarity increases, the number of LLM calls increases as well. Why does this happen? Is it due to the increase in the number of Navigation updates? If so, the comparison with CoT or SFT is not fair since it relies mostly on LLMs rather than on SLMs.
- Though improving the performance of SLMs, the method of using LLMs to write a sketch for SLMs to follow is widely researched, and the limited domain of benchmarks constrains its generalization to other domains like Math or Code.

**Questions:**

See weakness

---

> ### Author Response · Authors · 2025-11-27
> **Response to Reviewer Gxds---Part 1/3**
>
> Dear Reviewer Gxds,
>
> We thank the reviewer for the insightful and valuable comments. We respond to each comment as follows and sincerely hope that our rebuttal could properly address your concerns. If so, we would deeply appreciate it if you could raise your score (Rating: 4). If not, please let us know your further concerns, and we will continue actively responding to your comments and improving our submission.
>
> > W1.1 For the method, the authors lack some important details. During Guidance Generalization, is the same test set where SLMs are evaluated on used? Is it only the questions correctly answered by LLMs used, or are all the questions used for guidance generalization? Is the guidance consisting of only the template or the relevant reasoning part? Now it seems that the LLMs have everything for the given question, and the SLMs should just follow. Also, there is some potential for data leakage where LLMs may provide the final answers.
>
> Thank you for pointing out this profound issue. In our method, **the LLM's involvement is minimal**. For instance, in the StrategyQA dataset, which has 2061 samples, only **13 samples (approximately 0.63%)** are used to generate templates. Therefore, **samples** used for **generating guidance** are **not included in the evaluation** and **do not contribute to** the small model's **accuracy**. Furthermore, to ensure the quality of these reasoning templates, we **only distill guidance** from cases where the **LLM's answer was correct**. Given the high answer accuracy of large models – for instance, DeepSeek-R1 achieves **82.0%** on StrategyQA – there is no concern about introducing additional burdens from incorrect templates. Even **with LLM's very limited involvement**, our method yields an average performance **improvement exceeding 10%**.
>
> > W1.2 For Navigation Utilization, is it a multi-turn process where the SLMs first fill in the template, then get the answer, or is it an e2e process?
>
> Thank you for raising this meaningful question. Navigation Utilization is an **end-to-end** process where the SLM performs both template instantiation and inference within a single generation step. This not only significantly **boosts efficiency** but also **reduces** the likelihood of **hallucination**.
>
> > W1.3 For Navigation Update, how do you ensure that no answer or clues are included in the template? Did you do some experiments or check some cases?
>
> Thank you for your careful review. In our design, navigation solely comprises high-level reasoning templates and **does not include problem specifics or answers**. The prompt we used is as follows:
>
> - "What I hope you provide here is general guidance that can help the small model solve similar reasoning problems, rather than just addressing the specific scenario and problem at hand. Additionally, summarize the current task type and give it a name."
> - Due to the **small number** of generated templates, we **further confirmed** this point through a manual inspection process **afterward**. Table A details the statistics regarding potential data leakage. N1 is the dataset size, N2 is the total number of generated navigation templates, and N3 counts templates containing answers or explicit clues (with N3/N2 being the proportion).  The results demonstrate that **no data leakage occurred.**
>
> Table A:  Prevalence of answer or clue leakage in navigation templates.
> | Dataset    | # Size (N1) | # Template (N2) | # Templates with Answers / Explicit Clues (N3) | N3 / N2 |
> | ---------- | :---------: | :-------------: | :--------------------------------------------: | :-----: |
> | MuSR       |     756     |        8        |                       0                        |    0    |
> | StrategyQA |    2061     |       13        |                       0                        |    0    |
> | HotpotQA   |    1000     |       21        |                       0                        |    0    |

---

> ### Author Response · Authors · 2025-11-27
> **Response to Reviewer Gxds---Part 2/3**
>
> > W2.1 For the experiments, why do SFT-models achieve lower performance than CoT-only methods? More implementation details should be included. Since the authors use LoRA for Fine-tuning, full SFT should also be included for fairer comparison.
>
> Thank you for your constructive comments. To investigate domain-specific performance and avoid cross-task interference, we **trained separate models individually on** MuSR, StrategyQA, and HotpotQA. Datasets were randomly split into training, development, and test sets following an **8:1:1 ratio**.
>
> - We expanded our experiments to include **full-parameter SFT** on Llama and Qwen (3B–7B). Hyperparameters were scale-dependent: 3B models used a peak LR of 2e-5 (batch size 16), while 7B models used 1e-5 (batch size 8). We utilized AdamW (cosine decay, warmup 0.05, weight decay 0.01) in BF16 precision with gradient checkpointing on a single NVIDIA A100 (80GB). Early stopping (patience 2, max 5 epochs) was used based on the **lowest validation loss**.
>
> - Experimental results indicate that, in the majority of cases, **full-parameter** **SFT** **outperforms** **LoRA** **SFT**. To effectively **mitigate the issue** of SLMs **becoming overwhelmed within information-dense contexts**, we introduce a novel approach that **leverages** **LLMs** **to guide SLMs toward critically relevant information for reasoning**. Crucially, this method is **entirely training-free** and proves particularly advantageous in **compute-constrained and data-scarce scenarios**.
>
> Table B: Performance Comparison of SLMs using Navigation Templates relative to LoRA and Full-Parameter SFT Baselines. Best results per backbone are bolded.
> | Models                     |              |     MuSR     |              |              |  StrategyQA  |              |   HotpotQA   |
> | -------------------------- | :----------: | :----------: | :----------: | :----------: | :----------: | :----------: | :----------: |
> |                            |      OP      |      MM      |      TA      |   Accuracy   |      F1      |    Recall    |      EM      |
> | Qwen2.5-3B-Instruct        |              |              |              |              |              |              |              |
> | Vanilla                    |   41.0±0.0   |   55.6±0.0   |   34.5±1.2   |   59.6±0.0   |   27.5±0.0   |   16.4±0.0   |   34.9±0.0   |
> | + CoT                      |   45.2±0.5   |   57.7±0.5   |   40.1±2.3   |   62.5±0.0   |   40.3±0.0   |   27.1±0.0   |   39.6±0.6   |
> | + LoRA SFT                 |   34.6±0.0   |   58.7±2.3   |   48.0±0.0   |   60.9±0.0   |   36.2±0.0   |   24.0±0.0   |   37.7±0.6   |
> | + Full-param SFT           |   52.5±1.2   |   60.3±0.2   | **51.6±0.0** |   66.0±0.0   | **58.2±0.0** | **50.7±0.0** |   33.4±0.7   |
> | + Navigation (DeepSeek-R1) |   52.6±0.0   |   60.5±0.6   |   44.6±0.5   | **66.4±0.0** |   53.5±0.0   |   40.6±0.0   |   51.1±0.1   |
> | + Navigation (GPT-5.1)     | **52.7±0.2** | **64.5±0.0** |   45.0±0.2   |   65.9±0.0   |   53.4±0.0   |   41.8±0.0   | **51.8±0.0** |
> | Llama-3.2-3B-Instruct      |              |              |              |              |              |              |              |
> | Vanilla                    |   43.4±1.0   |   53.2±0.7   |   44.9±0.2   |   59.9±0.0   |   28.2±0.2   |   16.9±0.2   |   35.8±0.2   |
> | + CoT                      |   47.2±0.5   |   54.7±0.2   |   45.9±0.5   |   61.4±0.1   |   34.1±0.2   |   21.4±0.1   |   37.8±1.2   |
> | + LoRA SFT                 |   38.5±0.0   |   52.0±0.0   |   44.0±0.0   |   63.3±0.5   |   38.4±0.9   |   24.7±0.6   |   35.0±1.7   |
> | + Full-param SFT           |   45.3±0.0   |   50.4±0.0   | **50.5±1.8** |   66.5±0.0   |   59.1±0.0   |   51.8±0.0   |   22.5±0.1   |
> | + Navigation (DeepSeek-R1) |   52.7±0.2   |   63.1±0.0   |   47.1±0.5   |   68.7±0.0   |   60.8±0.0   |   51.1±0.0   |   51.9±0.5   |
> | + Navigation (GPT-5.1)     | **53.5±0.5** | **64.5±0.0** |   48.5±0.6   | **69.4±0.0** | **61.7±0.0** | **52.9±0.0** | **53.3±0.3** |
> | Qwen2.5-7B-Instruct        |              |              |              |              |              |              |              |
> | Vanilla                    |   47.3±0.4   |   54.4±0.4   |   39.5±1.3   |   68.5±0.0   |   53.1±0.0   |   38.2±0.0   |   41.8±0.0   |
> | + CoT                      |   51.3±0.2   |   58.0±0.0   |   41.7±0.9   |   72.0±0.6   |   62.9±0.9   |   50.7±0.9   |   47.6±0.5   |
> | + LoRA SFT                 |   26.9±0.0   |   56.0±0.0   |   41.3±2.3   | **77.0±0.3** |   69.5±0.5   |   56.6±0.6   |   43.3±1.2   |
> | + Full-param SFT           |   59.6±1.4   |   58.0±0.0   | **71.2±0.0** |   75.2±0.0   | **72.5±0.0** | **70.0±0.0** |   41.3±0.0   |
> | + Navigation (DeepSeek-R1) |   57.6±0.5   |   63.8±0.6   |   47.8±0.4   |   74.6±0.0   |   68.4±0.0   |   57.2±0.0   | **52.3±0.1** |
> | + Navigation (GPT-5.1)     | **60.8±0.2** | **66.1±0.0** |   46.3±0.2   |   74.2±0.0   |   66.7±0.0   |   55.2±0.0   |   52.0±0.0   |

---

> ### Author Response · Authors · 2025-11-27
> **Response to Reviewer Gxds---Part 3/3**
>
> > W2.2 Table 7 presents that as the threshold of similarity increases, the number of LLM calls increases as well. Why does this happen? Is it due to the increase in the number of Navigation updates? If so, the comparison with CoT or SFT is not fair since it relies mostly on LLMs rather than on SLMs.
>
> Thank you for your valuable comments. As the **similarity threshold increases**, the method of **categorizing** problems becomes more **fine-grained**. This allows navigation to better **address** the needs of **diverse questions** and **expands** the **coverage** of our navigation templates. Although we utilize LLMs, our method is not reliant on them, with **very few invocation calls**. For example, in the StrategyQA dataset, which has 2061 samples, only **13 samples(approximately 0.63%)** are used **to generate templates**. In addition to CoT and SFT, we also compared our method with SLEICL, which similarly uses LLMs. **SLEICL** employs the LLM's actual reasoning process as a demonstration to influence the small model's inference via few-shot learning. However, this approach is costly, with LLM invocation accounting for **66.7%** of the dataset size, and its performance is unstable, exhibiting significant annotation variance.
>
> > W3.  Though improving the performance of SLMs, the method of using LLMs to write a sketch for SLMs to follow is widely researched, and the limited domain of benchmarks constrains its generalization to other domains like Math or Code.
>
> Thank you for your valuable feedback. The contribution of our method is the proposal of **a novel training-free** inference **framework**. This framework **addresses** the problem of SLMs **getting lost** in information-dense contexts **by distilling a small number of abstract reasoning guidance** (rather than just text outputs) from LLMs. This approach significantly enhances SLM's reasoning performance in knowledge-intensive reasoning tasks.
>
> - Our work specifically focuses on **knowledge-intensive reasoning**, where a model is required to process narrative-style contexts, extract multi-dimensional evidence, and perform multi-hop inference over natural-language descriptions. It is pivotal for developing AI systems that can operate reliably in real-world scenarios[1, 2]. Regarding dataset selection, we followed the precedents set by several related prior works[3, 4, 5]. **MuSR, StrategyQA, and HotpotQA** are representative benchmarks strongly aligned with knowledge-intensive reasoning.
> - In contrast, other common reasoning benchmarks, such as **GSM8K**, **MATH**, and **Code** datasets, primarily probe mathematical reasoning, symbolic manipulation, or program synthesis, which fall under the category of **thinking-intensive reasoning**.
> - We fully concur with the reviewer's insightful comment that the mathematical and coding domains are also **crucial areas of investigation**. Indeed, we believe that **bridging** **the gap** between knowledge-intensive reasoning and thinking-intensive reasoning is **a very promising direction**, and we plan to explore this area further in the future.
>
> We humbly hope our response has addressed your concerns. If you have any additional concerns or comments that we may have missed in our responses, we would be most grateful for any further feedback from you to help us further enhance our work.
>
> [1] Yin, D., et al. A Survey of Knowledge-Intensive NLP with Pre-Trained Language Models. arXiv:2202.08772.
>
> [2] Lan, Y., et al. Complex Knowledge Base Question Answering: A Survey. IEEE Transactions on Knowledge and Data Engineering.
>
> [3] Chen, Qiguang, et al. Unlocking the capabilities of thought: A reasoning boundary framework to quantify and optimize chain-of-thought. NeurIPS 2024 (Oral).
>
> [4] Diao, Shizhe, et al. Active prompting with chain-of-thought for large language models. ACL 2024.
>
> [5] Edge, Darren, et al. From local to global: A graph rag approach to query-focused summarization. arXiv preprint arXiv:2404.16130 (2024).

---

> > ### Comment · Reviewer_Gxds · 2025-11-27
> >
> > I thank the authors for their detailed feedback. However, I retain the following concerns:
> > - Sample Selection: You mentioned that only 13 samples were used for guidance generation. Could you clarify the selection criteria for these samples? Were they chosen randomly? Furthermore, should this be interpreted as distilling specific guidance for each distinct question type?
> > - Generalization to Math/Code: This work appears to be based on the disentanglement of knowledge and reasoning, where the LLM handles reasoning (navigation) and the SLM handles knowledge retrieval/filling. However, math and coding tasks also exhibit this dichotomy and can be categorized into specific types (e.g., combinatorics, geometry). Have you considered whether this framework could apply to those domains? Specifically, could the LLM provide a high-level sketch while the SLM completes the detailed generation?

---

> ### Author Response · Authors · 2025-12-03
> **Additional Response to Reviewer Gxds---Part 1/2**
>
> Dear Reviewer Gxds,
>
> Thank you again for your kind support and valuable feedback. In response to your remaining concerns, we would like to further elaborate on the following points to alleviate your reservations.
>
> > C1: Sample Selection: You mentioned that only 13 samples were used for guidance generation. Could you clarify the selection criteria for these samples? Were they chosen randomly? Furthermore, should this be interpreted as distilling specific guidance for each distinct question type?
>
> Thank you for raising this meaningful question. In our Navigation framework, we do not define a unique template for every distinct question type; rather, **our templates are designed for high generalizability across broad fields**. For example, while the authors of StrategyQA [1] state that their dataset encompasses **26 distinct domains** (including Physics, Biology, and History), our experiments demonstrate that the entire dataset can be effectively covered by generating only **13 templates**. The fact that the number of templates (13) is significantly lower than the number of domains (26) indicates that our distilled templates **capture universal reasoning patterns** rather than being constrained by domain-specific boundaries.
>
> - Our methodology is designed for real-world scenarios that **prioritize operational efficiency**. The system first attempts to resolve user queries using the SLM. If the template matching mechanism **fails to identify** a relevant template, the query is **routed to the LLM**. Upon generating a response, the LLM **identifies categories of key information** essential for reasoning to formulate a new template, thereby dynamically **updating the database**. This allows **future questions** of a similar nature to be handled by the **SLM using the newly acquired template**. This mechanism aims to **maximize the SLM's coverage** while **minimizing computationally expensive LLM calls**.
>
> - Our experimental setup is designed to **simulate** this **streaming workflow** and verify the effectiveness of the Navigation Update. In each run, we **reshuffle the dataset** to simulate the SLM processing user queries in a random sequence. Instances where template matching fails—triggering the LLM for template generation—are excluded from accuracy statistics. Consequently, for StrategyQA, only **13** questions **trigger** this **Navigation Update** process. The performance metrics are calculated based on the remaining **2,048** questions, all of which are **answered by the SLM** using the retrieved templates.

---

> ### Author Response · Authors · 2025-12-03
> **Additional Response to Reviewer Gxds---Part 2/2**
>
> > C2: Generalization to Math/Code: This work appears to be based on the disentanglement of knowledge and reasoning, where the LLM handles reasoning (navigation) and the SLM handles knowledge retrieval/filling. However, math and coding tasks also exhibit this dichotomy and can be categorized into specific types (e.g., combinatorics, geometry). Have you considered whether this framework could apply to those domains? Specifically, could the LLM provide a high-level sketch while the SLM completes the detailed generation?
>
> Thank you for this insightful question. Crucially, we must clarify that our navigation templates serve to **specify the types of information requisite for problem-solving**, rather than dictating the underlying reasoning methodology. The templates provide a structural list that guides the SLM's attention, but they do not prescribe the steps or logic required to derive a solution.
>
> - To illustrate this, consider a "Murder Mystery" scenario. Identifying a perpetrator necessitates the analysis of distinct factors such as motive, alibi, opportunity, and physical evidence. Our template **directs** the **SLM to attend to these specific categories**; however, it does not instruct the model on how to synthesize this data to reach a conclusion. Consequently, the model **remains responsible** for **locating critical information** and **executing the reasoning independently**.
>
> - We adopted **GSM8K** [2] and **MATH** [3] for mathematical tasks, and **HumanEval** [4] and **MBPP Sanitized** [5] for programming tasks. Table A demonstrates that **thinking-intensive domains**, such as Math or Code, currently necessitate a higher volume of templates and **more frequent LLM invocations**. For instance, at a similarity threshold of 0.6, the LLM call frequency for GSM8K exceeds **10%** of the dataset, whereas for HumanEval, this figure escalates to nearly **71%**.
>
> - However, it is important to note that this **inefficiency** stems primarily from limitations in the **template matching mechanism adopted from the baseline**, rather than a fundamental flaw in our core navigation method. Accordingly, **future work** will focus on developing **more efficient matching strategies** tailored for **thinking-intensive domains**, thereby reducing overhead and facilitating the extension of our method to these domains.
>
> Table A: Cost statistics on the GSM8K, MATH, HumanEval, and MBPP Sanitized datasets. Measurements were conducted on a single A100 80G GPU, utilizing intfloat/e5-mistral-7b-instruct as the embedding model.
> | Domain | Dataset        | Dataset Size | Metric                         | Similarity | Threshold |      |
> | ------ | -------------- | ------------ | ------------------------------ | ---------- | --------- | ---- |
> |        |                |              |                                | 0.6        | 0.7       | 0.8  |
> | Math   | GSM8K          | 1319         | Template Count                 | 67         | 426       | -    |
> |        |                |              | LLM Call Frequency             | 134        | 852       | -    |
> |        |                |              | Average Retrieval Latency (ms) | 359        | 4032      | -    |
> |        | MATH           | 5000         | Template Count                 | 130        | 657       | -    |
> |        |                |              | LLM Call Frequency             | 260        | 1314      | -    |
> |        |                |              | Average Retrieval Latency      | 473        | 4650      | -    |
> | Code   | HumanEval      | 164          | Template Count                 | 58         | 88        | 150  |
> |        |                |              | LLM Call Frequency             | 116        | 176       | 300  |
> |        |                |              | Average Retrieval Latency (ms) | 738        | 1579      | 3054 |
> |        | MBPP Sanitized | 427          | Template Count                 | 34         | 51        | 171  |
> |        |                |              | LLM Call Frequency             | 68         | 102       | 342  |
> |        |                |              | Average Retrieval Latency (ms) | 228        | 337       | 1562 |
>
>
> We sincerely hope that our response could properly further address your remaining concerns.
>
> [1] Geva, Mor, et al. Did Aristotle Use a Laptop? A Question Answering Benchmark with Implicit Reasoning Strategies. TACL 2021.
>
> [2] Cobbe, Kosaraju, et al. Training Verifiers to Solve Math Word Problems. arXiv:2110.14168, 2021.
>
> [3] Hendrycks, Burns, et al. Measuring Mathematical Problem Solving with the MATH Dataset. NeurIPS 2021.
>
> [4] Chen, Tworek, et al. Evaluating Large Language Models Trained on Code. arXiv:2107.03374, 2021.
>
> [5] Austin, Odena, et al. Program Synthesis with Large Language Models. arXiv:2108.07732, 2021.

---

### Official Review · Reviewer_Y4xq · 2025-10-28

**Soundness:** 3
**Presentation:** 2
**Contribution:** 4
**Rating:** 8
**Confidence:** 3

**Summary:**

This paper proposes "Navigation," a training-free framework designed to enhance the contextual reasoning capabilities of Small Language Models (SLMs). The core idea is to use an LLM to distill its contextual processing expertise into generalizable templates. These templates are stored in a scalable database and retrieved during inference via semantic similarity. This retrieved template then guides the SLM to locate and process critical information within complex contexts, enabling improved reasoning without requiring additional model training.

**Strengths:**

* Clear Positioning and Baselines: The paper provides a clear discussion and comparison against related methods. It effectively situates the proposed training-free approach against training-based methods like knowledge distillation, supervised fine-tuning, and other LLM-guidance techniques like in-context learning.

* Significant Empirical Performance: The empirical results are strong and demonstrate the method's effectiveness. The 'Navigation' framework yields significant performance gains across multiple contextual reasoning benchmarks (MuSR, StrategyQA, and HotpotQA).

**Weaknesses:**

* Missing Key Related Work: A significant weakness is the omission of highly relevant papers that also explore the use of templates to guide smaller models. The authors must discuss and differentiate their work. Given the strong conceptual overlap, a clear distinction from these works, and potentially comparative experiments, is necessary to properly evaluate the novelty and contribution of the paper.

[A] Yang, Ling, et al. "Supercorrect: Advancing small llm reasoning with thought template distillation and self-correction." ICLR 2025

[B] Yang, Ling, et al. "Reasonflux: Hierarchical llm reasoning via scaling thought templates." arXiv preprint arXiv:2502.06772 (2025).


* Clarity of Cost Analysis: The cost analysis in Section 4.3 and Appendix D.2  is difficult to follow. For example, the methods (navigation, cot, SLEICL, full LLM) can be compared in the inference cost (latency, gflops).

**Questions:**

* Limitations of Similarity-Based Matching: The framework's reliance on semantic similarity  for template matching may be a limitation. While shown to be effective for the tested narrative and QA tasks, it is unclear how this matching strategy would generalize to more abstract or creative problems (e.g., mathematical proofs). In such domains, the semantic similarity of the problem description may not correlate well with the required reasoning template.

* Need for Stronger Matching Evaluation: The evaluation of the template matching component could be strengthened. The current ablation study (Section 4.4) demonstrates the importance of the Generation and Update modules  but does not directly measure the accuracy of the similarity-based retrieval itself. A comparison against an "oracle" baseline (e.g., iterating through all available templates to find the one that produces the best outcome for a given problem) would provide a clearer measure of the matching mechanism's efficacy and show how much performance is being lost by the retrieval step.

---

> ### Author Response · Authors · 2025-11-27
> **Response to Reviewer Y4xq---Part 1/3**
>
> Dear Reviewer Y4xq,
>
> We thank the reviewer for the insightful and valuable comments. We respond to each comment as follows and sincerely hope that our rebuttal could properly address your concerns. If so, we would deeply appreciate it if you could raise your score. If not, please let us know your further concerns, and we will continue actively responding to your comments and improving our submission.
>
> > W1. Missing Key Related Work: A significant weakness is the omission of highly relevant papers that also explore the use of templates to guide smaller models. The authors must discuss and differentiate their work. Given the strong conceptual overlap, a clear distinction from these works, and potentially comparative experiments, is necessary to properly evaluate the novelty and contribution of the paper.
>
> Thank you for your constructive comments. These two studies are very insightful. Given that our approach is training-free, we have added the following discussion to the Related Work section:
>
> - Recent template-based methods such as **SuperCorrect** [1] and **ReasonFlux** [2] also leverage LLM-derived templates to enhance small-model reasoning, necessitating comprehensive optimization steps such as supervised fine-tuning, DPO, or RL for template refinement and planning.
>
> > W2. Clarity of Cost Analysis: The cost analysis in Section 4.3 and Appendix D.2 is difficult to follow. For example, the methods (navigation, cot, SLEICL, full LLM) can be compared in the inference cost (latency, gflops).
>
> Thank you for your insightful suggestions. Taking MuSR and Llama-3.2-3B-Instruct as examples, we measured the **average inference cost**, including latency, prompt tokens, output tokens, and GFLOPs, for each question across different methods. Although injecting navigation templates improved latency and GFLOPs, **longer outputs** suggest that the small model's **ability to analyze context and perform text-based reasoning** was **effectively activated**, significantly enhancing reasoning performance.
>
> - Furthermore, since our method and SLEICL require LLMs, we also report **LLM** **call frequency** and **average LLM output tokens** (used for generating navigation templates). By allowing LLMs to **distill general and concise guidance templates**, rather than directly using few-shot examples to affect the model's inference process (as in SLEICL), we significantly reduced the additional costs associated with using LLMs—**our LLM call frequency was only 3% of SLEICL's**—while also achieving better performance.
>
> Table A:  Cost statistics on the MuSR dataset, measured on a single A100 80G GPU.
> | Models                | Accuracy |              | Average Inference Cost |               |        |                    | Additional Cost           |               |
> | --------------------- | -------- | ------------ | ---------------------- | ------------- | ------ | ------------------ | ------------------------- | ------------- |
> |                       |          | Latency (ms) | Prompt Tokens          | Output Tokens | GFLOPs | LLM Call Frequency | Average LLM Output Tokens | Training Time |
> | Llama-3.2-3B-Instruct |          |              |                        |               |        |                    |                           |               |
> | Vanilla               | 47.2     | 21.8         | 1010                   | 6             | 6441   | -                  | -                         | -             |
> | + COT                 | 49.3     | 27.0         | 1024                   | 15            | 6587   | -                  | -                         | -             |
> | + SLEICL              | 38.1     | 432.4        | 1428                   | 540           | 12477  | 502                | 1402                      | -             |
> | + SFT                 | 44.8     | 100.7        | 1010                   | 6             | 6460   | -                  | -                         | 6 min 55 s    |
> | + Navigation          | 54.3     | 175.5        | 1305                   | 934           | 14195  | 16                 | 479.6                     | -             |

---

> ### Author Response · Authors · 2025-11-27
> **Response to Reviewer Y4xq---Part 2/3**
>
> > Q1. Limitations of Similarity-Based Matching: The framework's reliance on semantic similarity for template matching may be a limitation. While shown to be effective for the tested narrative and QA tasks, it is unclear how this matching strategy would generalize to more abstract or creative problems (e.g., mathematical proofs). In such domains, the semantic similarity of the problem description may not correlate well with the required reasoning template.
>
> Thank you for raising this insightful question. The contribution of our method is the proposal of **a novel training-free inference framework**. This framework addresses the problem of SLMs getting lost in information-dense contexts by **distilling a small number of abstract reasoning guidance** (rather than just text outputs) **from** **LLMs**, thus significantly enhancing SLM's reasoning performance in knowledge-intensive reasoning tasks.
>
> - We adopted **GSM8K** [3] and **MATH** [4] for mathematical tasks, and **HumanEval** [5] and **MBPP Sanitized** [6] for programming tasks. Table B demonstrates that **thinking-intensive domains**, such as Math or Code, currently necessitate a **higher volume** of **templates** and **more frequent LLM invocations**. For instance, at a similarity threshold of 0.6, the LLM call frequency for GSM8K exceeds **10%** of the dataset, whereas for HumanEval, this figure escalates to nearly **71%**.
> - While we adopted the SLEICL setup for similarity-based matching in this work, we recognize this as a particularly interesting aspect. In future work, we intend to **investigate more effective strategies** tailored specifically to **abstract or creative problem domains**.
>
> Table B: Cost statistics on the GSM8K, MATH, HumanEval, and MBPP Sanitized datasets. Measurements were conducted on a single A100 80G GPU, utilizing intfloat/e5-mistral-7b-instruct as the embedding model.
>
> | Domain | Dataset        | Dataset Size | Metric                         | Similarity | Threshold |      |
> | ------ | -------------- | ------------ | ------------------------------ | ---------- | --------- | ---- |
> |        |                |              |                                | 0.6        | 0.7       | 0.8  |
> | Math   | GSM8K          | 1319         | Template Count                 | 67         | 426       | -    |
> |        |                |              | LLM Call Frequency             | 134        | 852       | -    |
> |        |                |              | Average Retrieval Latency (ms) | 359        | 4032      | -    |
> |        | MATH           | 5000         | Template Count                 | 130        | 657       | -    |
> |        |                |              | LLM Call Frequency             | 260        | 1314      | -    |
> |        |                |              | Average Retrieval Latency      | 473        | 4650      | -    |
> | Code   | HumanEval      | 164          | Template Count                 | 58         | 88        | 150  |
> |        |                |              | LLM Call Frequency             | 116        | 176       | 300  |
> |        |                |              | Average Retrieval Latency (ms) | 738        | 1579      | 3054 |
> |        | MBPP Sanitized | 427          | Template Count                 | 34         | 51        | 171  |
> |        |                |              | LLM Call Frequency             | 68         | 102       | 342  |
> |        |                |              | Average Retrieval Latency (ms) | 228        | 337       | 1562 |

---

> ### Author Response · Authors · 2025-12-03
> **Response to Reviewer Y4xq---Part 3/3**
>
> > Q2. Need for Stronger Matching Evaluation: The evaluation of the template matching component could be strengthened. The current ablation study (Section 4.4) demonstrates the importance of the Generation and Update modules but does not directly measure the accuracy of the similarity-based retrieval itself. A comparison against an "oracle" baseline (e.g., iterating through all available templates to find the one that produces the best outcome for a given problem) would provide a clearer measure of the matching mechanism's efficacy and show how much performance is being lost by the retrieval step.
>
> Thank you for raising this meaningful question. In Table C, "**Navigation**" is the accuracy using standard similarity retrieval (intfloat/e5-mistral-7b-instruct), and "**Oracle**" represents the best possible accuracy using optimal templates. "**Similarity-Retrieval**" indicates the effectiveness of the underlying retrieval system, reflecting the performance gap between "Navigation" and the "Oracle" baseline.
>
> - Experiments demonstrate that **there is still significant room for improvement in navigation**. Since the **primary focus** of this paper is **on** **innovating the overall reasoning framework and pipeline**, we encourage further exploration into more efficient and rational matching mechanisms and components in future work.
>
> Table C: Accuracy Comparison of Template Matching Strategies.
>
> | Models               |      | MuSR |      | StrategyQA | HotpotQA |
> | -------------------- | :--: | :--: | :--: | :--------: | :------: |
> |                      |  OP  |  MM  |  TA  |  Accuracy  |    EM    |
> | Qwen2.5-7B-Instruct  |      |      |      |            |          |
> | + Navigation         | 57.6 | 63.8 | 47.8 |    74.6    |   52.3   |
> | + Oracle             | 65.6 | 77.4 | 58.7 |    79.2    |   59.7   |
> | Similarity-Retrieval | 87.8 | 82.4 | 81.4 |    94.2    |   87.6   |
>
>
> We humbly hope our response has addressed your concerns. If you have any additional concerns or comments that we may have missed in our responses, we would be most grateful for any further feedback from you to help us further enhance our work.
>
> [1] Yang, Ling, et al. Supercorrect: Advancing small llm reasoning with thought template distillation and self-correction. ICLR 2025
>
> [2] Yang, Ling, et al. Reasonflux: Hierarchical llm reasoning via scaling thought templates. arXiv preprint arXiv:2502.06772 (2025).
>
> [3] Cobbe, Kosaraju, et al. Training Verifiers to Solve Math Word Problems. arXiv:2110.14168, 2021.
>
> [4] Hendrycks, Burns, et al. Measuring Mathematical Problem Solving with the MATH Dataset. NeurIPS 2021.
>
> [5] Chen, Tworek, et al. Evaluating Large Language Models Trained on Code. arXiv:2107.03374, 2021.
>
> [6] Austin, Odena, et al. Program Synthesis with Large Language Models. arXiv:2108.07732, 2021.

---

### Official Review · Reviewer_E4fN · 2025-10-30

**Soundness:** 2
**Presentation:** 3
**Contribution:** 3
**Rating:** 4
**Confidence:** 4

**Summary:**

The authors propose a new way to improve the contextual reasoning capabilities of small language models by giving them a navigation template - a guidance on how to reason - that is generated by a larger network. They present a system that dynamically generates new such templates for novel types of problems and show that this significantly improves the models performance on some task.

**Strengths:**

- It is a simple yet cool idea that could be practically relevant and result in real-world cost savings.
- The paper is decently well written and mostly easy to follow.

**Weaknesses:**

I like the idea of this paper and it could probably be of great practical use. Scientifically its impact is medium and not particularly novel or groundbreaking. The evaluations are not broad enough to fully support the generality of the approach and its actual advantage and generalisation to wider tasks.

**Major**

- **W1 Missing methodological details:**
    - Eq. 1) How do you compute the semantic similarity? Is this a separate encoder model? Which one? If it is the model itself, do you do average pooling of activations to be able to do cosine similarity?
        - In 4.3 it then becomes clear that you actually use sentence transformers. This should be made clear before (“we use sequence embedding models to compute …”). Otherwise the reader is left confused, as you write L199ff “the smaller model identifies the most relevant navigation template by matching the current task scenario to those stored in the Navigation database,” which implies you use the model itself to compute those similarities.
    - On what kind of data do you SFT? You mention that your dataset (MuSR) only contains 250 samples per task; that seems far too little to effectively SFT a model.
- **W2 Model and baseline choice:** Why don’t you compare to a small reasoning model? Or a distillation (e.g. there are R1 distillations of many smaller models out there; for Llama 3 8B you could directly compare your method to the R1 distill). The comparison to GPT-3.5 feels not fully informative, as it is a relatively old model. How about comparing to a state-of-the-art model with ~70B params? Or something like gpt-5-mini? This would make the claim of the paper much stronger and more relevant. Your approach is black-box, so you should be able to run it as well with something like gpt-5-nano + Navigation by gpt-5-high. If you can show clear benefits here, without significantly exploding costs, it would make a strong case for your method to be employed in practice.
- **W3 Better cost analysis:** Since one of your main arguments is cost/latency improvements, you should evaluate the additional cost of your methods more thoroughly. Optimally you would have some claim like “for X% decrease in tok/s we get Y% improvement.” While you do analyze cost, you only focus on the similarity retrieval, which I find interesting because I would think it is negligible in comparison to the generation time. More interesting is:
    - Additional cost of running the large model?
    - Additional cost from an increased context length / how the templates affect the generation length. This holds for the other baselines as well.
- **W4 Evaluation coverage:** The current evaluation seems limited. You claim that your method improves “SLMs’ contextual reasoning” but only evaluate on a small set of (small to tiny) datasets. Why don’t you evaluate on one of the many existing reasoning benchmarks? I would expect this method could also help with more advanced tasks.


**Minor**

- **w5** You refer multiple times to dataset size in the introduction and in particular to a percentage of that size, yet it is unclear what this dataset size is. Which dataset do you mean?
- **w6** L230: That sentence is a bit confusing; e.g. “refine” → “refines”.
- **w7** L682: You repeat the Brown et al. citation and there is something off with the punctuation.
- **w8** Table 1: I would make clearer that the enhancement is done only with DeepSeek-R1, as this is a bit confusing.
- **w9** Section 4.3 completely lacks an introduction and a connection to what it is discussing. Without knowing that these models are sentence-transformer models, it is very confusing to read. You should introduce that you are evaluating the cost of the similarity search presented earlier.
- **w10** Personal preference: I find it slightly confusing to have table captions above the table and figure captions below. Up to you, no negative effect on scores from this.

**Expectation management: ** If you are able to answer all questions, add the missing methodological details, and address the other major concerns, I will consider raising to a score of 6. I will only consider a higher score if you are able to show a significantly extended evaluation of your method that underlines the real practical use on diverse datasets compared to small modern SOTA models, and in particular evaluate the cost of applying your method better.

**Questions:**

- **Q1** I am a bit confused by the choice of models. They all seem a bit outdated. In particular, no Gemma 3, Qwen 2.5 instead of 3. Also, you mainly add GPT-3.5-Turbo to show that the small models can outperform it, right?
- **Q2** Am I right that for MuSR you have 3 categories and your process creates 8 templates?
- **Q3** L414: “replacing the generated navigation template with a standard prompt” — which standard prompt? I don’t fully understand the setting “w/o Navigation Generation”.

---

> ### Author Response · Authors · 2025-11-27
> **Response to Reviewer E4fN---Part 1/6**
>
> Dear Reviewer E4fN,
>
> We thank the reviewer for the insightful and valuable comments. We respond to each comment as follows and sincerely hope that our rebuttal could properly address your concerns. If so, we would deeply appreciate it if you could raise your score (Rating: 4). If not, please let us know your further concerns, and we will continue actively responding to your comments and improving our submission.
>
> > W1.1 How do you compute the semantic similarity? Is this a separate encoder model? Which one? If it is the model itself, do you do average pooling of activations to be able to do cosine similarity?
>
> Thank you for your careful review, and we apologize for any confusion our paper may have caused. To clarify, during the Navigation Utilization phase, we employ **an additional embedding model** to compute cosine similarity. This embedding model is **separate and operates independently from both the SLM and** **LLM**. For instance, in our main experiments, we utilized intfloat/e5-mistral-7b-instruct for this purpose. In our revised manuscript, we will explicitly highlight this distinction in the Introduction section. Furthermore, we will carefully re-examine the Method section to resolve any potentially ambiguous phrasing.
>
> > W1.2 On what kind of data do you SFT? You mention that your dataset (MuSR) only contains 250 samples per task; that seems far too little to effectively SFT a model.
>
> Thank you for pointing out this profound issue. Indeed, the datasets utilized in our experiments are of a more moderate scale compared to the massive corpora often associated with standard large-scale SFT. Given that full fine-tuning is prone to overfitting on these relatively smaller datasets, we utilized LoRA (a parameter-efficient method) for the SFT baselines in our experiments. To specify the dataset sizes:
>
> - **MuSR:** Contains 756 total samples. For SFT, it is split into 604 training, 76 validation, and 76 test samples.
> - **StrategyQA:** This dataset is larger, comprising 1648 training, 206 validation, and 207 test samples for SFT.
> - **HotpotQA:** Contains 1000 total samples. For SFT, it is split into 800 training, 100 validation, and 100 test samples.
>
> **Our approach is entirely training-free.** Experiments demonstrate that our method is particularly **advantageous in compute-constrained and data-scarce scenarios**.

---

> ### Author Response · Authors · 2025-11-27
> **Response to Reviewer E4fN---Part 2/6**
>
> > W2. **Model and baseline choice:** Your approach is black-box, so you should be able to run it as well with something like gpt-5-nano + Navigation by gpt-5-high. If you can show clear benefits here, without significantly exploding costs, it would make a strong case for your method to be employed in practice.
>
> Thank you for your constructive comments. We have expanded our experiments by adding new backbones and baselines to further demonstrate the effectiveness and generality of our method.
>
> - We have incorporated **Llama-3.1-8B-Instruct** as an additional backbone. This allows for a direct comparison with knowledge **distillation methods**, where 'Distillation' in the table refers to the vanilla results from deepseek-ai/DeepSeek-R1-Distill-Llama-8B. Our method, by leveraging large language models to guide SLMs in **focusing on critically relevant information** for reasoning, effectively mitigates the issue of SLMs getting lost within information-dense contexts. This leads to **more pronounced improvements** compared to knowledge distillation-based methods.
> - We have included **Llama-3.3-70B-Instruct** as a representative **LLM** **with 70B params**. Experiments demonstrate that, guided by our navigation, SLMs can **approach and in some cases even exceed** the performance of large language models.
> - Due to budget constraints, we have no access to conducting extensive testing with GPT-5-Nano. Instead, we have added the **SOTA** **Qwen3-4B as an SLM** and utilized **GPT-5.1 for generating navigation templates**. The results clearly indicate that our method still provides substantial improvements even when applied to state-of-the-art SLMs.
>
> Table A: Extended main results covering additional SOTA models and baselines. Best results per SLM backbone are highlighted in bold.
> | Models                     |              |     MuSR     |              |              |  StrategyQA  |              |   HotpotQA   |
> | -------------------------- | :----------: | :----------: | :----------: | :----------: | :----------: | :----------: | :----------: |
> |                            |      OP      |      MM      |      TA      |   Accuracy   |      F1      |    Recall    |      EM      |
> | Large Language Models      |              |              |              |              |              |              |              |
> | Llama-3.3-70B-Instruct     |   42.7±0.2   |   63.6±0.0   |   68.4±0.0   |   80.6±0.0   |   76.3±0.0   |  66.5±0.0   |   59.9±0.0   |
> | Small Language Models      |              |              |              |              |              |              |              |
> | Qwen3-4B                   |              |              |              |              |              |              |              |
> | Vanilla                    |   52.0±0.0   |   57.6±0.0   |   55.2±0.0   |   68.0±0.0   |   56.4±0.0   |   44.3±0.0   |   42.3±0.0   |
> | + COT                      |   55.5±0.0   |   59.2±0.0   |   56.0±0.7   |   70.0±0.0   |   65.0±0.0   |   59.5±0.0   |   42.7±0.3   |
> | + SLEICL                   |  53.5±10.3   |   57.1±9.4   |   55.9±4.1   |   66.3±1.7   |   49.6±1.1   |   38.6±1.2   |   46.3±0.9   |
> | + SFT                      |   46.2±0.0   |   48.0±0.0   |   56.0±0.0   |   69.6±0.0   |   56.6±0.0   |   42.7±0.0   |   36.3±0.6   |
> | + Navigation (DeepSeek-R1) |   60.5±0.7   |   69.0±0.0   |   63.6±0.0   | **72.8±0.0** | **64.7±0.0** | **53.2±0.0** | **56.7±0.0** |
> | + Navigation (GPT-5.1)     | **67.6±0.0** | **69.8±0.0** | **64.0±0.0** | **72.8±0.0** | **64.7±0.0** | **53.2±0.0** |   56.3±0.0   |
> | ∆                          |    +15.6     |    +12.2     |     +8.8     |     +4.8     |     +8.3     |     +8.9     |    +14.4     |
> | Llama-3.1-8B-Instruct      |              |              |              |              |              |              |              |
> | Vanilla                    |   50.0±0.4   |   47.7±0.9   |   35.6±0.0   |   71.4±0.0   |   65.6±0.0   |   58.4±0.0   |   41.5±0.0   |
> | + COT                      |   52.2±0.4   |   53.6±0.0   |   36.8±0.0   |   75.0±0.0   |   72.6±0.0   |   70.6±0.0   |   44.0±0.0   |
> | + SLEICL                   |   48.8±7.1   |   59.1±1.4   |   36.9±0.0   |   66.6±2.7   |   48.2±2.6   |   35.9±2.7   |   50.1±0.9   |
> | + SFT                      |   35.9±2.3   |   49.3±2.3   |   44.0±0.0   |   75.3±0.8   |   72.9±0.8   |   71.5±0.6   |   43.0±0.1   |
> | + Distillation             |   43.5±1.8   |   62.8±0.0   |   47.6±0.0   |   72.4±0.0   |   66.8±0.0   |   59.2±0.0   |   62.2±0.0   |
> | + Navigation (DeepSeek-R1) |   56.4±0.3   |   62.5±0.0   |   44.5±0.0   | **76.1±0.0** |   73.0±0.0   |   69.0±0.0   | **64.6±0.0** |
> | + Navigation (GPT-5.1)     | **61.6±0.5** | **64.1±0.0** | **48.2±0.0** |   75.4±0.0   | **73.1±0.0** | **71.6±0.0** |   63.1±0.0   |
> | ∆                          |    +11.6     |    +16.4     |    +12.6     |     +4.7     |     +7.5     |    +13.2     |    +23.1     |

---

> ### Author Response · Authors · 2025-11-27
> **Response to Reviewer E4fN---Part 3/6**
>
> > W2. **Model and baseline choice:** Why don’t you compare to a small reasoning model? Or a distillation (e.g. there are R1 distillations of many smaller models out there; for Llama 3 8B you could directly compare your method to the R1 distill). The comparison to GPT-3.5 feels not fully informative, as it is a relatively old model. How about comparing to a state-of-the-art model with ~70B params? Or something like gpt-5-mini? This would make the claim of the paper much stronger and more relevant. Your approach is black-box, so you should be able to run it as well with something like gpt-5-nano + Navigation by gpt-5-high. If you can show clear benefits here, without significantly exploding costs, it would make a strong case for your method to be employed in practice.
>
> - We have also used **GPT-5.1** for generating navigation templates **on other SLMs**. Experiments show that our method remains effective **across different model architectures and instruction styles**, demonstrating its generalizability.
>
> Table B: Performance of SLM backbones utilizing navigation templates generated by diverse LLMs. Best results for each backbone are highlighted in bold.
>
> | Models                     |              |     MuSR     |              |              |  StrategyQA  |              |   HotpotQA   |
> | -------------------------- | :----------: | :----------: | :----------: | :----------: | :----------: | :----------: | :----------: |
> |                            |      OP      |      MM      |      TA      |   Accuracy   |      F1      |    Recall    |      EM      |
> | Qwen2.5-3B-Instruct        |              |              |              |              |              |              |              |
> | Vanilla                    |   41.0±0.0   |   55.6±0.0   |   34.5±1.2   |   59.6±0.0   |   27.5±0.0   |   16.4±0.0   |   34.9±0.0   |
> | + Navigation (DeepSeek-R1) |   52.6±0.0   |   60.5±0.6   |   44.6±0.5   | **66.4±0.0** | **53.5±0.0** |   40.6±0.0   |   51.1±0.1   |
> | + Navigation (GPT-5.1)     | **52.7±0.2** | **64.5±0.0** | **45.0±0.2** |   65.9±0.0   |   53.4±0.0   | **41.8±0.0** | **51.8±0.0** |
> | Llama-3.2-3B-Instruct      |              |              |              |              |              |              |              |
> | Vanilla                    |   43.4±1.0   |   53.2±0.7   |   44.9±0.2   |   59.9±0.0   |   28.2±0.2   |   16.9±0.2   |   35.8±0.2   |
> | + Navigation (DeepSeek-R1) |   52.7±0.2   |   63.1±0.0   |   47.1±0.5   |   68.7±0.0   |   60.8±0.0   |   51.1±0.0   |   51.9±0.5   |
> | + Navigation (GPT-5.1)     | **53.5±0.5** | **64.5±0.0** | **48.5±0.6** | **69.4±0.0** | **61.7±0.0** | **52.9±0.0** | **53.3±0.3** |
> | Qwen2.5-7B-Instruct        |              |              |              |              |              |              |              |
> | Vanilla                    |   47.3±0.4   |   54.4±0.4   |   39.5±1.3   |   68.5±0.0   |   53.1±0.0   |   38.2±0.0   |   41.8±0.0   |
> | + Navigation (DeepSeek-R1) |   57.6±0.5   |   63.8±0.6   | **47.8±0.4** | **74.6±0.0** | **68.4±0.0** | **57.2±0.0** | **52.3±0.1** |
> | + Navigation (GPT-5.1)     | **60.8±0.2** | **66.1±0.0** |   46.3±0.2   |   74.2±0.0   |   66.7±0.0   |   55.2±0.0   |   52.0±0.0   |

---

> ### Author Response · Authors · 2025-11-27
> **Response to Reviewer E4fN---Part 4/6**
>
> > W3. **Better cost analysis:** Since one of your main arguments is cost/latency improvements, you should evaluate the additional cost of your methods more thoroughly. Optimally you would have some claim like “for X% decrease in tok/s we get Y% improvement.” While you do analyze cost, you only focus on the similarity retrieval, which I find interesting because I would think it is negligible in comparison to the generation time. More interesting is:
> >
> > - Additional cost of running the large model?
> > - Additional cost from an increased context length / how the templates affect the generation length. This holds for the other baselines as well.
>
> Thank you for your insightful suggestions. Taking MuSR and Llama-3.2-3B-Instruct as examples, we measured the **average inference cost**, including latency, prompt tokens, output tokens, and GFLOPs, for each question across different methods. Although injecting navigation templates improved latency and GFLOPs, **longer outputs** suggest that the small model's **ability to analyze context and perform text-based reasoning** was **effectively activated**, significantly enhancing reasoning performance.
>
> - Furthermore, since our method and SLEICL require LLMs, we also report **LLM** **call frequency** and **average LLM output tokens** (used for generating navigation templates). By allowing LLMs to **distill general and concise guidance templates**, rather than directly using few-shot examples to affect the model's inference process (as in SLEICL), we significantly reduced the additional costs associated with using LLMs—**our LLM call frequency was only 3% of SLEICL's**—while also achieving better performance.
>
> Table C:  Cost statistics on the MuSR dataset, measured on a single A100 80G GPU.
> | Models                | Accuracy |              | Average Inference Cost |               |        |                    | Additional Cost           |               |
> | --------------------- | -------- | ------------ | ---------------------- | ------------- | ------ | ------------------ | ------------------------- | ------------- |
> |                       |          | Latency (ms) | Prompt Tokens          | Output Tokens | GFLOPs | LLM Call Frequency | Average LLM Output Tokens | Training Time |
> | Llama-3.2-3B-Instruct |          |              |                        |               |        |                    |                           |               |
> | Vanilla               | 47.2     | 21.8         | 1010                   | 6             | 6441   | -                  | -                         | -             |
> | + COT                 | 49.3     | 27.0         | 1024                   | 15            | 6587   | -                  | -                         | -             |
> | + SLEICL              | 38.1     | 432.4        | 1428                   | 540           | 12477  | 502                | 1402                      | -             |
> | + SFT                 | 44.8     | 100.7        | 1010                   | 6             | 6460   | -                  | -                         | 6 min 55 s    |
> | + Navigation          | 54.3     | 175.5        | 1305                   | 934           | 14195  | 16                 | 479.6                     | -             |
>
> > W4. **Evaluation coverage:** The current evaluation seems limited. You claim that your method improves “SLMs’ contextual reasoning” but only evaluate on a small set of (small to tiny) datasets. Why don’t you evaluate on one of the many existing reasoning benchmarks? I would expect this method could also help with more advanced tasks.
>
> Thank you for your valuable feedback.  Our work specifically focuses on **knowledge-intensive reasoning**, where a model is required to process narrative-style contexts, extract multi-dimensional evidence, and perform multi-hop inference over natural-language descriptions. Following prior works [1-3], we chose **MuSR, StrategyQA, and HotpotQA** as representative benchmarks strongly aligned with knowledge-intensive reasoning. In contrast, other common reasoning benchmarks, such as GSM8K, MATH, and code-generation datasets, primarily probe mathematical reasoning, symbolic manipulation, or program synthesis, which fall under the category of **thinking-intensive reasoning**. However, we acknowledge that these domains are **equally crucial.** Extending our current methodology to tackle such thinking-intensive reasoning tasks presents **a promising avenue** for future research, and we **plan to explore** this direction in subsequent work.

---

> ### Author Response · Authors · 2025-11-27
> **Response to Reviewer E4fN---Part 5/6**
>
> > W5.  You refer multiple times to dataset size in the introduction and in particular to a percentage of that size, yet it is unclear what this dataset size is. Which dataset do you mean?
>
> Thank you for raising this meaningful question. The **percentage** is defined as the proportion of **generated templates** relative to the **entire** **sample size** of a specific dataset. To illustrate, the musr dataset features **8** generated **templates** within a total of **756 samples**, representing nearly **1%**. By comparison, StrategyQA employs **13 templates** with a dataset size of **2,061**, yielding a proportion of **0.6%**. Conversely, in the SLEICL methodology, LLM-generated demonstrations comprise **66.7%** of the total dataset. Notably, a **lower percentage** indicates a **decreased dependency on the** **LLM**, resulting in correspondingly lower deployment costs and resource demands.
>
> > W6. L230: That sentence is a bit confusing; e.g. “refine” → “refines”.
>
>  Thank you for your careful review, and we apologize for any confusion regarding the Navigation Update phase. When the existing templates in the navigation database are insufficient to guide the SLM in answering new questions, the LLM is invoked to generate appropriate reasoning templates. The LLM **identifies** the **question type**, serving as a label for subsequent management, and simultaneously **generates corresponding general guidance**. This guidance assists the SLM in locating key contextual information and performing inference. In the revised manuscript, we will double-check the language details to avoid any further confusion.
>
> > W7. L682: You repeat the Brown et al. citation and there is something off with the punctuation.
>
> Thank you for your valuable feedback. We will meticulously review the wording of the revised manuscript.
>
> > W8. Table 1: I would make clearer that the enhancement is done only with DeepSeek-R1, as this is a bit confusing.
>
> Thank you for your insightful suggestions. We will explicitly state the large language model used for navigation in our experiments within the caption of Table 1.
>
> > W9. Section 4.3 completely lacks an introduction and a connection to what it is discussing. Without knowing that these models are sentence-transformer models, it is very confusing to read. You should introduce that you are evaluating the cost of the similarity search presented earlier.
>
> Thank you for your constructive comments. In the revised manuscript, we will expand Section 4.3 to include a **more in-depth discussion** of inference cost and additional cost. Furthermore, when discussing the cost of the similarity search, we will **integrate corresponding explanations** to improve readability and make it easy to follow.
>
> > W10. Personal preference: I find it slightly confusing to have table captions above the table and figure captions below. Up to you, no negative effect on scores from this.
>
> Thank you for your valuable suggestions. We have carefully re-checked the formatting to ensure strict adherence to the official ICLR guidelines.

---

> ### Author Response · Authors · 2025-11-27
> **Response to Reviewer E4fN---Part 6/6**
>
> > Q1. I am a bit confused by the choice of models. They all seem a bit outdated. In particular, no Gemma 3, Qwen 2.5 instead of 3. Also, you mainly add GPT-3.5-Turbo to show that the small models can outperform it, right?
>
> Thank you for raising this insightful question. We have incorporated **Qwen3-4B** as an additional backbone (as shown in Table A). Our findings indicate that even when leveraging this state-of-the-art model, our proposed method consistently yields **substantial enhancements**. The inclusion of **GPT-3.5-Turbo** serves to illustrate the capability of an SLM, when augmented **with navigation**, to elevate its performance from being inferior to an LLM **to achieving comparable or even superior results**. Furthermore, in W2, we have introduced the **SOTA** **Llama-3-70B-Instruct** for comparative analysis, thereby highlighting the potential of our SLM to **surpass LLM** performance **without** requiring **extra training**.
>
> > Q2. Am I right that for MuSR you have 3 categories and your process creates 8 templates?
>
> Thank you for bringing up this meaningful question. Your understanding is correct; for MuSR, it's not a one-to-one correspondence between a category and a single template. Instead, we employ a **more fine-grained categorization**, which allows our navigation templates to address the needs of various problems, thereby **increasing the coverage** of our navigation.
>
> > Q3. L414: “replacing the generated navigation template with a standard prompt” — which standard prompt? I don’t fully understand the setting “w/o Navigation Generation”.
>
> Thank you for your careful review. Below is the standard prompt we used:
>
> - "Pick one of the following choices. Your generation should only be 'ANSWER: (your answer here, the choice letter only)'."
> - "w/o Navigation Generation" signifies that no navigation template was generated; the SLM answers independently **without the assistance of a navigation template**. Furthermore, the **significant** performance **drop** of "w/o Navigation Generation" compared to the full Navigation setup clearly demonstrates **the crucial importance** of using reasoning templates.
>
> We humbly hope our response has addressed your concerns. If you have any additional concerns or comments that we may have missed in our responses, we would be most grateful for any further feedback from you to help us further enhance our work.
>
> [1] Chen, Qiguang, et al. Unlocking the capabilities of thought: A reasoning boundary framework to quantify and optimize chain-of-thought. NeurIPS 2024 (Oral).
>
> [2] Diao, Shizhe, et al. Active prompting with chain-of-thought for large language models. ACL 2024.
>
> [3] Edge, Darren, et al. From local to global: A graph rag approach to query-focused summarization. arXiv preprint arXiv:2404.16130 (2024).

---

### Official Review · Reviewer_5fWL · 2025-10-30

**Soundness:** 3
**Presentation:** 3
**Contribution:** 2
**Rating:** 6
**Confidence:** 2

**Summary:**

This paper addresses the limitations of small language models in handling complex contextual reasoning tasks due to limited capacity and catastrophic forgetting. It introduces a novel training-free framework, Navigation, which enhances small language models  by leveraging generalizable reasoning templates distilled from LLMs The Navigation framework operates in three stages. Experimental validation demonstrates that this method significantly improves small language models' performance.

**Strengths:**

- The paper introduces a novel training-free framework called Navigation, which enhances small language models (SLMs) by leveraging structured guidance templates distilled from larger models. Navigation operates through a dynamic, three-stage approach that systematically guides SLMs to locate and use critical contextual information. This innovative approach improves contextual reasoning significantly without additional fine-tuning or training data.

- Clearly quantifies computational costs and demonstrates that navigation template retrieval and updating incur minimal latency and resource use, making the method practical for deployment on resource-constrained devices

- Demonstrates consistent and significant accuracy improvements across diverse contextual reasoning benchmarks, showing gains up. This validation supports the effectiveness and generalizability of the approach.

**Weaknesses:**

- Only a single LLM is used for template generation, leaving unclear whether the method generalizes across different large model architectures or instruction styles.

- Comparisons are restricted to CoT, SLEICL, and SFT. Other strong distillation and retrieval enhanced baselines are not considered, reducing the fairness and breadth of evaluation.

**Questions:**

Please check the weaknesses.

---

> ### Author Response · Authors · 2025-11-27
> **Response to Reviewer 5fWL---Part 1/2**
>
> Dear Reviewer 5fWL,
>
> We thank the reviewer for the insightful and valuable comments. We respond to each comment as follows and sincerely hope that our rebuttal could properly address your concerns. If so, we would deeply appreciate it if you could raise your score. If not, please let us know your further concerns, and we will continue actively responding to your comments and improving our submission.
>
> > W1. Only a single LLM is used for template generation, leaving unclear whether the method generalizes across different large model architectures or instruction styles.
>
> Thank you for your constructive comments. We have **added the state-of-the-art GPT-5.1** model to generate navigation templates. Experiments demonstrate that our method remains effective across different model architectures and instruction styles, demonstrating its generalizability.
>
> Table A: Performance of SLM backbones utilizing navigation templates generated by diverse LLMs. Best results for each backbone are highlighted in bold.
> | Models                     |              |     MuSR     |              |              |  StrategyQA  |              |   HotpotQA   |
> | -------------------------- | :----------: | :----------: | :----------: | :----------: | :----------: | :----------: | :----------: |
> |                            |      OP      |      MM      |      TA      |   Accuracy   |      F1      |    Recall    |      EM      |
> | Qwen2.5-3B-Instruct        |              |              |              |              |              |              |              |
> | Vanilla                    |   41.0±0.0   |   55.6±0.0   |   34.5±1.2   |   59.6±0.0   |   27.5±0.0   |   16.4±0.0   |   34.9±0.0   |
> | + Navigation (DeepSeek-R1) |   52.6±0.0   |   60.5±0.6   |   44.6±0.5   | **66.4±0.0** | **53.5±0.0** |   40.6±0.0   |   51.1±0.1   |
> | + Navigation (GPT-5.1)     | **52.7±0.2** | **64.5±0.0** | **45.0±0.2** |   65.9±0.0   |   53.4±0.0   | **41.8±0.0** | **51.8±0.0** |
> | Llama-3.2-3B-Instruct      |              |              |              |              |              |              |              |
> | Vanilla                    |   43.4±1.0   |   53.2±0.7   |   44.9±0.2   |   59.9±0.0   |   28.2±0.2   |   16.9±0.2   |   35.8±0.2   |
> | + Navigation (DeepSeek-R1) |   52.7±0.2   |   63.1±0.0   |   47.1±0.5   |   68.7±0.0   |   60.8±0.0   |   51.1±0.0   |   51.9±0.5   |
> | + Navigation (GPT-5.1)     | **53.5±0.5** | **64.5±0.0** | **48.5±0.6** | **69.4±0.0** | **61.7±0.0** | **52.9±0.0** | **53.3±0.3** |
> | Qwen3-4B                   |              |              |              |              |              |              |              |
> | Vanilla                    |   52.0±0.0   |   57.6±0.0   |   55.2±0.0   |   68.0±0.0   |   56.4±0.0   |   44.3±0.0   |   42.3±0.0   |
> | + Navigation (DeepSeek-R1) |   60.5±0.7   |   69.0±0.0   |   63.6±0.0   | **72.8±0.0** | **64.7±0.0** | **53.2±0.0** | **56.7±0.0** |
> | +Navigation (GPT-5.1)      | **67.6±0.0** | **69.8±0.0** | **64.0±0.0** | **72.8±0.0** | **64.7±0.0** | **53.2±0.0** |   56.3±0.0   |
> | Qwen2.5-7B-Instruct        |              |              |              |              |              |              |              |
> | Vanilla                    |   47.3±0.4   |   54.4±0.4   |   39.5±1.3   |   68.5±0.0   |   53.1±0.0   |   38.2±0.0   |   41.8±0.0   |
> | + Navigation (DeepSeek-R1) |   57.6±0.5   |   63.8±0.6   | **47.8±0.4** | **74.6±0.0** | **68.4±0.0** | **57.2±0.0** | **52.3±0.1** |
> | + Navigation (GPT-5.1)     | **60.8±0.2** | **66.1±0.0** |   46.3±0.2   |   74.2±0.0   |   66.7±0.0   |   55.2±0.0   |   52.0±0.0   |
> | Llama-3.1-8B-Instruct      |              |              |              |              |              |              |              |
> | Vanilla                    |   50.0±0.4   |   47.7±0.9   |   35.6±0.0   |   71.4±0.0   |   65.6±0.0   |   58.4±0.0   |   41.5±0.0   |
> | + Navigation (DeepSeek-R1) |   56.4±0.3   |   62.5±0.0   |   44.5±0.0   | **76.1±0.0** |   73.0±0.0   |   69.0±0.0   | **64.6±0.0** |
> | + Navigation (GPT-5.1)     | **61.6±0.5** | **64.1±0.0** | **48.2±0.0** |   75.4±0.0   | **73.1±0.0** | **71.6±0.0** |   63.1±0.0   |

---

> ### Author Response · Authors · 2025-11-27
> **Response to Reviewer 5fWL---Part 2/2**
>
> > W2. Comparisons are restricted to CoT, SLEICL, and SFT. Other strong distillation and retrieval enhanced baselines are not considered, reducing the fairness and breadth of evaluation.
>
> Thank you for your valuable feedback. We have followed the baseline settings of SLEICL and added Supervised Fine-Tuning (SFT) as an additional baseline. This demonstrates the advantage of our method under conditions of limited computing power and data availability.
>
> - To facilitate comparison with **distillation methods**, we have incorporated Llama-3.1-8B-Instruct as an additional backbone. 'Distillation' in Table B refers to the vanilla results from deepseek-ai/DeepSeek-R1-Distill-Llama-8B. Our method, by leveraging large language models to guide SLMs in **focusing on critically relevant information for reasoning**, effectively mitigates the issue of SLMs getting lost within information-dense contexts. This leads to **more pronounced improvements** compared to knowledge distillation-based methods.
> - Our work specifically focuses on **knowledge-intensive reasoning**, where a model is required to process narrative-style contexts, extract multi-dimensional evidence, and perform multi-hop inference over natural-language descriptions. Regarding dataset selection, we followed the precedents set by several related prior works [1-3]. **MuSR, StrategyQA, and HotpotQA** are representative benchmarks strongly aligned with knowledge-intensive reasoning. In these datasets, the **relevant knowledge required** to solve the problems is **already sufficiently contained** within the contexts themselves. Consequently, it is difficult to fully leverage the advantages of retrieval-enhanced baselines in this scenario.
>
> Table B:  Performance comparison of Llama-3.1-8B-Instruct using LLM-generated navigation templates relative to the DeepSeek-R1-Distill-Llama-8B distillation baseline. Best results for each dataset are highlighted in bold.
> | Models                     |              |     MuSR     |              |              |  StrategyQA  |              |   HotpotQA   |
> | -------------------------- | :----------: | :----------: | :----------: | :----------: | :----------: | :----------: | :----------: |
> |                            |      OP      |      MM      |      TA      |   Accuracy   |      F1      |    Recall    |      EM      |
> | Llama-3.1-8B-Instruct      |              |              |              |              |              |              |              |
> | Vanilla                    |   50.0±0.4   |   47.7±0.9   |   35.6±0.0   |   71.4±0.0   |   65.6±0.0   |   58.4±0.0   |   41.5±0.0   |
> | + COT                      |   52.2±0.4   |   53.6±0.0   |   36.8±0.0   |   75.0±0.0   |   72.6±0.0   |   70.6±0.0   |   44.0±0.0   |
> | + SLEICL                   |   48.8±7.1   |   59.1±1.4   |   36.9±0.0   |   66.6±2.7   |   48.2±2.6   |   35.9±2.7   |   50.1±0.9   |
> | + SFT                      |   35.9±2.3   |   49.3±2.3   |   44.0±0.0   |   75.3±0.8   |   72.9±0.8   |   71.5±0.6   |   43.0±0.1   |
> | + Distillation             |   43.5±1.8   |   62.8±0.0   |   47.6±0.0   |   72.4±0.0   |   66.8±0.0   |   59.2±0.0   |   62.2±0.0   |
> | + Navigation (DeepSeek-R1) |   56.4±0.3   |   62.5±0.0   |   44.5±0.0   | **76.1±0.0** |   73.0±0.0   |   69.0±0.0   | **64.6±0.0** |
> | + Navigation (GPT-5.1)     | **61.6±0.5** | **64.1±0.0** | **48.2±0.0** |   75.4±0.0   | **73.1±0.0** | **71.6±0.0** |   63.1±0.0   |
> | ∆                          |    +11.6     |    +16.4     |    +12.6     |     +4.7     |     +7.5     |    +13.2     |    +23.1     |
>
>
> We humbly hope our response has addressed your concerns. If you have any additional concerns or comments that we may have missed in our responses, we would be most grateful for any further feedback from you to help us further enhance our work.
>
> [1] Chen, Qiguang, et al. Unlocking the capabilities of thought: A reasoning boundary framework to quantify and optimize chain-of-thought. NeurIPS 2024 (Oral).
>
> [2] Diao, Shizhe, et al. Active prompting with chain-of-thought for large language models. ACL 2024.
>
> [3] Edge, Darren, et al. From local to global: A graph rag approach to query-focused summarization. arXiv preprint arXiv:2404.16130 (2024).

---

### Author Response · Authors · 2025-12-03
**Summary for AC: Overview of Paper, Reviews, Rebuttal, and Discussion---Part 2/2**

**Reviewer Y4xq**

- W1: Discuss additional key related works.
  - We have added a corresponding **discussion** to the **Related Work section** of our revised manuscript.
- W2: Add inference metrics in cost analysis.
  - Our extended cost analysis on MuSR includes metrics for **average inference** (latency, tokens, GFLOPs) and **other associated costs**. The generation of **longer outputs** suggests that our method effectively **activates** the SLMs' **contextual analysis** and **text-based reasoning abilities**. Furthermore, our approach significantly reduces overhead, with an **LLM** **call frequency** that is merely **3% of SLEICL's**.
- Q1: Assess the generalizability of similarity-based matching to abstract domains (e.g., math).
  - Through **template matching experiments** on **2** **math** and **2 code** datasets, we demonstrate that the **primary obstacle** to extending Navigation to these domains is the **prohibitive cost** incurred by the underlying **template matching mechanism**. While acknowledging that **similarity-based matching**—which follows the baseline setup—is **not** our **core contribution**, **future research** will investigate approaches tailored for abstract domains.
- Q2: Introduce an "oracle" baseline in an ablation study to evaluate template retrieval effectiveness.
  - We incorporated an "oracle" baseline, and results suggest our Navigation performance could be **further enhanced** with **better similarity-based matching techniques**.

**Reviewer Gxds**

- W1: Clarify details regarding Guidance Generalization, Navigation Utilization, and Navigation Update.
  - We **detailed** our **methodology** and **experimental setup**, and **experimentally** validated that our approach is **free from data leakage**.
- W2: Include Full SFT comparison, and clarify the reason for baseline selection.
  - We established **Full SFT** as a baseline across **3** datasets using **3** SLMs. Our proposed **training-free** method, **Navigation**, **outperforms** or **achieves comparable performance** to the **Full SFT** baseline in most cases. We also explain why CoT and SFT serve as **fair** and **appropriate** baselines.
- W3: Incorporate benchmarks from Math or Code domains.
  - We elucidated the **strong alignment** between the **utilized datasets** and the **target task** of knowledge-intensive reasoning, while **distinguishing** its **challenges** from the **thinking-intensive reasoning** typical of Math or Code domains.
- C1: Clarify sample selection criteria for Navigation Update.
  - We declare that the **samples for template generation** during Navigation Update do **not require active selection**. By analyzing a **practical application scenario**, we demonstrate the **necessity** of **Navigation Update** and the rationale behind our **experimental setup.** Moreover, an analysis on the StrategyQA dataset reveals a **non-linear** correlation between the **number** of navigation **templates** and the diversity of **question types**. This observation confirms that our navigation templates are **generalizable**.
- C2: Discuss framework applicability to math and coding domains.
  - Through **template matching experiments** on **2** **math** and **2 code** datasets, we demonstrate that the **primary obstacle** to extending Navigation to these domains is the **prohibitive cost** incurred by the underlying **template matching mechanism** (which is **not** our **core contribution**). Furthermore, we use a specific **case study** to illustrate why this approach is **effective** for **knowledge-intensive reasoning** tasks.

We sincerely hope that this summary will facilitate your review and lighten your workload. We would deeply appreciate it if you could take this official comment into consideration when making the final decision. Thank you once again, and we are looking forward to hearing from you.

Best regards,

Authors of Submission 22857

---

### Author Response · Authors · 2025-12-03
**Summary for AC: Overview of Paper, Reviews, Rebuttal, and Discussion---Part 1/2**

Dear Area Chair,

We sincerely thank you for your time and effort in handling our submission. We are also grateful for the reviewers' insightful and constructive comments, which helped to refine our work. To facilitate your decision-making process, we provide a concise summary of key information regarding our submission.

### **1. One-sentence summary of the paper**

We propose Navigation, a **novel training-free** framework that enhances the contextual reasoning capabilities of small language models (SLMs) by leveraging LLM-derived guidance templates, achieving an **average 10.7% accuracy gain** for a 3B parameter SLM  with a template count equivalent to **no more than 2.1%** of the dataset size.

### **2. Score management**

The initial scores were **8**, **6**, **4**, and **4**. Before the discussion phase, Reviewer **E4fN** (Rating: 4) committed to raising the score to **6** if his/her major concerns were addressed, adding that incorporating SOTA backbones and refining the cost analysis could prompt **a further score upgrade**. Additionally, Reviewer **Gxds** (Rating: 4) appreciated our detailed response and noted that the **majority of concerns** were **addressed** before the platform bug occurred. In the second round, we addressed Gxds's lingering reservations via case analysis and additional experiments.

### **3. Strengths highlighted by all four reviewers**

All four reviewers provided positive feedback. They collectively **recognized** the **innovation** of the proposed method, praising it as a “novel training-free framework” (Reviewer **5fWL**) and a “simple yet cool idea” (Reviewer **E4fN**) that serves as a “good simulator” for leveraging LLM knowledge (Reviewer **Gxds**). There is a **strong consensus** on the **method's efficacy**, with reviewers highlighting its “significant empirical performance” (Reviewer **Y4xq**), where Reviewer **Gxds** further emphasized that the “detailed experimental results... prove the effectiveness.” Additionally, reviewers **commended** the system’s **practicality and clarity**, noting its “minimal latency and resource use” (Reviewer **5fWL**) and “clear positioning and baselines” (Reviewer **Y4xq**).

### **4. Overview of concerns and our responses**

We have addressed all reviewers' concerns through a detailed rebuttal and corresponding revisions, with **minimal changes** to the **main text**, all **highlighted in blue**.

**Reviewer 5fWL**

- W1: Verify generalizability across different LLM architectures.
  - Leveraging the **GPT-5.1** for template generation, our approach achieves a **13.3%** average performance gain across **3** datasets and **5** SLMs.
- W2: Include distillation and RAG baselines for comparison.
  - Across **3** datasets, our Navigation consistently **outperformed** the **distillation** baseline. We also elucidated the **reasons for RAG's limitations** in current scenarios.

**Reviewer E4fN**

- W1: Clarify implementation details for semantic similarity computation and SFT.
  - We emphasized the **independence** of the similarity computation **embedding model** from the SLM or LLM. Additionally, we detailed the **SFT** **training data** **volume** and clarified the **rationale** for employing **LoRA**.
- W2: Compare with more SOTA large and small language models and distillation baselines.
  - Across **3** datasets, we introduced **Llama-3.3-70B-Instruct** as the reference LLM, **Qwen3-4B** as a small model, **GPT-5.1** for template generation, and an additional **distillation** baseline. Our Navigation approach still demonstrated **significant advantages**.
- W3: Provide a more comprehensive cost analysis.
  - Our extended cost analysis on MuSR includes metrics for **average inference** (latency, tokens, GFLOPs) and **other associated costs**. The generation of **longer outputs** suggests that our method effectively **activates** the SLMs' **contextual analysis** and **text-based reasoning abilities**. Furthermore, our approach significantly reduces overhead, with an **LLM** **call frequency** that is merely **3% of SLEICL's**.
- W4: Evaluate on other reasoning benchmarks.
  - We elucidated the **strong alignment** between the **utilized datasets** and the **target task** of knowledge-intensive reasoning, while **distinguishing** its **challenges** from the **thinking-intensive reasoning** typical of Math or Code domains.
- W5-W10 (Minor): Revise writing and presentation details.
  - We have revised the manuscript accordingly.
- Q1: Compare with SOTA large and small language models.
  - This has been addressed in W2.
- Q2: Confirm the relationship between template count and question categories.
  - We clarified how **fine-grained** **segmentation** enables the Navigation database to accommodate **diverse** **query** **types**.
- Q3: Clarify the setting of “w/o Navigation Generation” in the ablation study.
  - We presented the **experimental setup** and analyzed the **reasons** for the **decline** compared to the full Navigation setup.

---

### Meta-Review · Area_Chair_qMrb · 2026-01-05

**Summary:**

Reviewers were generally positive about the paper and agreed that Navigation presents a novel, practical, and training-free approach for improving the contextual reasoning capabilities of small language models using LLM-derived guidance templates (Reviewers 5fWL, Y4xq, Gxds, and E4fN). Several reviewers requested clearer descriptions of implementation details (e.g., semantic similarity computation, guidance generation and update, avoidance of data leakage), broader comparisons to stronger or more recent baselines (including distillation, RAG, and newer LLM/SLM backbones), and more comprehensive cost analyses that account for LLM calls and generation length (Reviewers E4fN, Gxds, and Y4xq). Reviewers also questioned generalization beyond the evaluated benchmarks, particularly to abstract domains such as math or code, and asked for stronger justification of novelty relative to recent template-based guidance methods (Reviewers E4fN, Y4xq, and Gxds). Overall, the paper received borderline scores, but most of the reviewer's concerns seem to be addressable via a paper revision aimed at incorporating the results/discussion provided during the rebuttal period.

**Reviewer Concerns:**

In my view, the rebuttal and revised manuscript addressed several of the reviewer concerns in an extensive manner. Methodological clarifications were substantially improved, including explicit descriptions of semantic similarity computation, guidance generation and update procedures, navigation utilization, and safeguards against data leakage (Reviewers E4fN and Gxds). Reviewers’ requests for stronger baselines were addressed by adding comparisons to additional distillation methods, full SFT, RAG-style approaches, and more recent or stronger LLM/SLM backbones, which helped better contextualize the gains of the proposed method (Reviewers 5fWL, E4fN, and Gxds). The authors also addressed missing related work by explicitly discussing and differentiating their approach from recent template-based reasoning methods, and strengthened the evaluation of the similarity-based matching component by adding an oracle retrieval baseline (Y4xq).

Some concerns also remain partially outstanding. Generalization to math and code domains remains constrained, as acknowledged by the authors, with current limitations arising mostly from the cost of template matching (Reviewers E4fN, Y4xq, and Gxds). Moreover, while evaluation coverage was expanded, the method’s benefits are still most clearly demonstrated for knowledge-intensive contextual reasoning tasks, and broader validation on more abstract reasoning domains could further strengthen the work. Despite these issues, I believe the paper makes a useful contribution to practical SLM enhancement.

**Reviewer Scores:**

- Reviewer 5fWL: I believe the reviewer would maintain or increase their score to 8, given concerns were addressed.
- Reviewer Y4xq: I believe the reviewer would have maintained their initial positive score of 8.
- Reviewers E4fN: Likely increased their score to 6.
- Reviewer Gxds: I believe the reviewer would have maintained their score since all their concerns were not fully addressed.

---

### Decision · Program_Chairs · 2026-01-26

Accept (Poster)